# Cellular metabolism constrains innate immune responses in early human ontogeny

Bernard Kan [1,2], Christina Michalski [1,2], Helen Fu[1,2], Hilda H.T. Au [3], Kelsey Lee [1,2], Elizabeth A. Marchant[1,2], Maye F. Cheng[1,2], Emily Anderson-Baucum[4], Michal Aharoni-Simon[1,9], Peter Tilley[5,6], Raghavendra G. Mirmira[4], Colin J. Ross [1,7], Dan S. Luciani[1,8], Eric Jan [3] & Pascal M. Lavoie [1,2]

Pathogen immune responses are profoundly attenuated in fetuses and premature infants, yet the mechanisms underlying this developmental immaturity remain unclear. Here we show transcriptomic, metabolic and polysome profiling and find that monocytes isolated from infants born early in gestation display perturbations in PPAR-γ-regulated metabolic pathways, limited glycolytic capacity and reduced ribosomal activity. These metabolic changes are linked to a lack of translation of most cytokines and of MALT1 signalosome genes essential to respond to the neonatal pathogen *Candida*. In contrast, they have little impact on house-keeping phagocytosis functions. Transcriptome analyses further indicate a role for mTOR and its putative negative regulator *DNA Damage Inducible Transcript 4-Like* in regulating these metabolic constraints. Our results provide a molecular basis for the broad susceptibility to multiple pathogens in these infants, and suggest that the fetal immune system is metabolically programmed to avoid energetically costly, dispensable and potentially harmful immune responses during ontogeny.

[1] BC Children's Hospital Research Institute, 950 West 28th Avenue, Vancouver, BC V5Z 4H4, Canada. [2] Department of Pediatrics, University of British Columbia, Vancouver, BC V6H 3V4, Canada. [3] Department of Biochemistry and Molecular Biology, Life Sciences Institute, University of British Columbia, 5457-2350 Health Sciences Mall, Vancouver, BC V6T 1Z3, Canada. [4] Departments of Medicine and Pediatrics and the Center for Diabetes and Metabolic Diseases, Indiana University School of Medicine, 1044 West Walnut Street, Indianapolis, IN 46202, USA. [5] BC Children's and Women's Hospitals, 4480 Oak St, Vancouver, BC V6H 3N1, Canada. [6] Department of Pathology & Laboratory Medicine, University of British Columbia, Vancouver, BC V6T 2B5, Canada. [7] Faculty of Pharmaceutical Sciences, University of British Columbia, 2405 Wesbrook Mall, Vancouver, BC V6T 1Z3, Canada. [8] Department of Surgery, University of British Columbia, Vancouver, BC V5Z 1M9, Canada. [9] Present address: Ophthalmology Research Lab, Kaplan Medical Center, Rehovot 76100, Israel. These authors contributed equally: Bernard Kan, Christina Michalski. Correspondence and requests for materials should be addressed to P.M.L. (email: plavoie@cw.bc.ca)

Infection is a leading causes of neonatal mortality worldwide, responsible for over one million deaths each year in infants under 28 days of age[1]. Infants born prematurely are particularly vulnerable to severe infections. Indeed, one out of six preterm infants born below 37 weeks of gestation will develop a life-threatening infection in their first month or life, owing to immaturity of their immune defenses[2,3].

*Candida species (spp.)* are a major neonatal pathogen, and remain an important cause of mortality from sepsis in premature infants despite a decreasing incidence of candidiasis over the last decade[4]. In healthy adults, this micro-organism rarely causes invasive disease[5,6]. In contrast, newborns are more vulnerable, especially those who are born below 33 weeks of gestation[4]. This warrants research to understand the molecular basis for their increased susceptibility to *Candida*, but also to other neonatal pathogens[7]. Prevention of systemic *Candida* infection in humans requires immune recognition and phagocytosis via Pattern Recognition Receptors (PRR)[8]. PRR-mediated immune responses are profoundly attenuated during gestation, until about 29 to 33 weeks[9,10]. Despite the major clinical impact of these functional deficits, we lack a molecular understanding of how these responses are regulated during ontogeny.

*Candida spp.* can exist as yeast or hyphae[5]. C-type lectins and Toll-like receptors (TLRs) are the main PRRs involved in the immune recognition of *Candida spp*[8]. Yeast forms are strongly detected by the C-type lectin receptor dectin-1 and predominate in the bloodstream during invasion. In contrast, hyphae can also be recognized via TLR2 and TLR4 in addition to a main role for C-type lectin receptors[11–14]. Dectin-1 binds the fungal cell wall component β-1,3-glucan[15,16], resulting in the production of pro-inflammatory cytokines including interleukin-1β (IL-1β). This cytokine is crucial for rapid innate immune responses[17], but also for the generation of a long-lasting mucosal immunity against *Candida*[18]. At the cellular level, monocytes (in blood) and macrophages (in tissues) are the main source of IL-1β. Dendritic cells play an important role in presenting fungal antigens to T cells. However, monocytes demonstrate increased reactivity to *Candida albicans* compared to dendritic cells and macrophages, highlighting the importance of the former in preventing disseminated infection through blood[19].

Production of IL-1β in monocytes can occur via two main cellular pathways: In the canonical pathway, activation of PRRs results in expression of the *Il1b* gene, which is then translated into the pro-IL-1β precursor protein until a second danger-associated signal (e.g., tissue damage) leads to proteolytic cleavage of pro-IL-1β and secretion of its mature IL-1β form via the NLRP3/Caspase-1 inflammasome (reviewed in ref. [8]). Alternatively, in the non-canonical pathway, both the production of pro-IL-1β and its cleavage into mature IL-1β by caspase-8 occur via activation of the MALT1/Bcl10/CARD9 signalosome pathway[20,21]. In humans, the importance of the signalosome is evidenced by data showing increased invasive *Candida* infections in subjects carrying loss-of-function mutations in CARD9[22].

Despite decades of research into the functional characterization of the neonatal immune system, we lack an understanding of the mechanisms regulating PRR responsiveness during ontogeny[2,3,23]. To address this question, we studied responses to *Candida spp.*, a major neonatal pathogen that is recognized through multiple PRRs, and for which relatively little is known in human newborns. Combining transcriptomic, metabolic and polysome profiling studies in an unbiased way, we find that neonatal monocytes are metabolically skewed and lack translation of key immune response genes in a gestational age-dependent manner. Our data provide a mechanism whereby the broad attenuation in PRR responses in early gestation can concurrently increase vulnerability of preterm infants to multiple neonatal

pathogens. In light of these results, we propose that preterm immune cells are metabolically reprogrammed to avoid innate immune activation signals in early ontogeny, at the same time offering potential therapeutic avenues to restore immune deficits and reduce infections in these high-risk infants.

## Results

**Lack of anti-fungal immune recognition in early gestation**. To examine responses to *Candida species*, we first compared monocytes' ability to phagocytose clinical strains of this micro-organism between preterm, term neonatal and adult subjects. Notably, preterm cells phagocytosed *Candida albicans* as well as adults (Fig. 1a). On the other hand, mononuclear cells from preterm neonates were unable to produce a cytokine response in presence of *Candida albicans* or *C. parapsilosis*, as demonstrated by a lack of production of IL-1β, IL-6 (Fig. 1b, c) and of other cytokines (Supplementary Figure 1). When assaying specific receptors, preterm neonatal cells also did not respond to dectin-1 (using curdlan), TLR2 (using zymosan) or TLR4 (using LPS) stimulation (Fig. 1d–g) despite a strong response detected in adult and term neonatal cells. At the transcript level, responses to LPS were generally stronger then responses to curdlan across all three age groups (Supplementary Figure 2). However, cytokine gene responses to curdlan were reduced in preterm subjects, suggesting an upstream dectin-1 signaling deficit in the latter age group (Supplementary Figure 2).

The importance of dectin-1 for recognition of *Candida* was confirmed by efficient blocking of cytokine responses using a neutralizing receptor antibody (Fig. 2a). On the other hand, dectin-1 blocking only partially inhibited phagocytosis of this pathogen (Fig. 2b). Moreover, phagocytosis of *Candida* was unaffected by blocking other receptors known to be important for the uptake of this micro-organism (Fig. 2c, d), which is consistent with a functional redundancy of these receptors[8]. Altogether, our data suggest a functional impairment in the PRR mediating the immune recognition of *Candida* in preterm monocytes, thereby abrogating cytokine responses, but not phagocytosis of this pathogen.

**Broad metabolic impairments in preterm monocytes**. The lack of cytokine response to *Candida* in preterm cells was concerning given the major importance of this pathogen mediating invasive disease in these infants. Yet, the molecular basis for this vulnerability is unknown. To investigate this, we used an unbiased, systems approach comparing the transcriptome of adult, term and preterm monocytes, at the genome-wide level. First, a gene ontology analysis comparing the transcriptome of unstimulated preterm, term neonatal and adult monocytes revealed major differences located mainly in pathways involved in glycolysis, oxidative phosphorylation and beta-oxidation metabolism (Fig. 3a–c; Supplementary Figure 3; Supplementary Figure 4). Notably, differences were seen also with a profound downregulation of ribosomal genes (Fig. 3d) and an over-representation of down-regulated genes involved in translation initiation (adjusted $p = 8.3 \times 10^{-21}$), translation (adj. $p = 2.6 \times 10^{-07}$), and cell activation pathways (adj. $p = 2.6 \times 10^{-16}$; Supplementary Data 1).

In light of the reduced responses to curdlan in preterm cells, we opted to compare gene expression responses to LPS instead. Upon stimulation with LPS, a large proportion of genes (~40%) were comparatively upregulated in all 3 age groups (Fig. 4a, b), including strong gene expression of *Il1b*, *Tnfa*, and *Il6* (Fig. 4c), as well as other cytokine/chemokine genes (Supplementary Figure 5). In independent qPCR experiments, expression of the *IL1b* and *IL6* cytokine genes was detectable after 30 min, and followed a

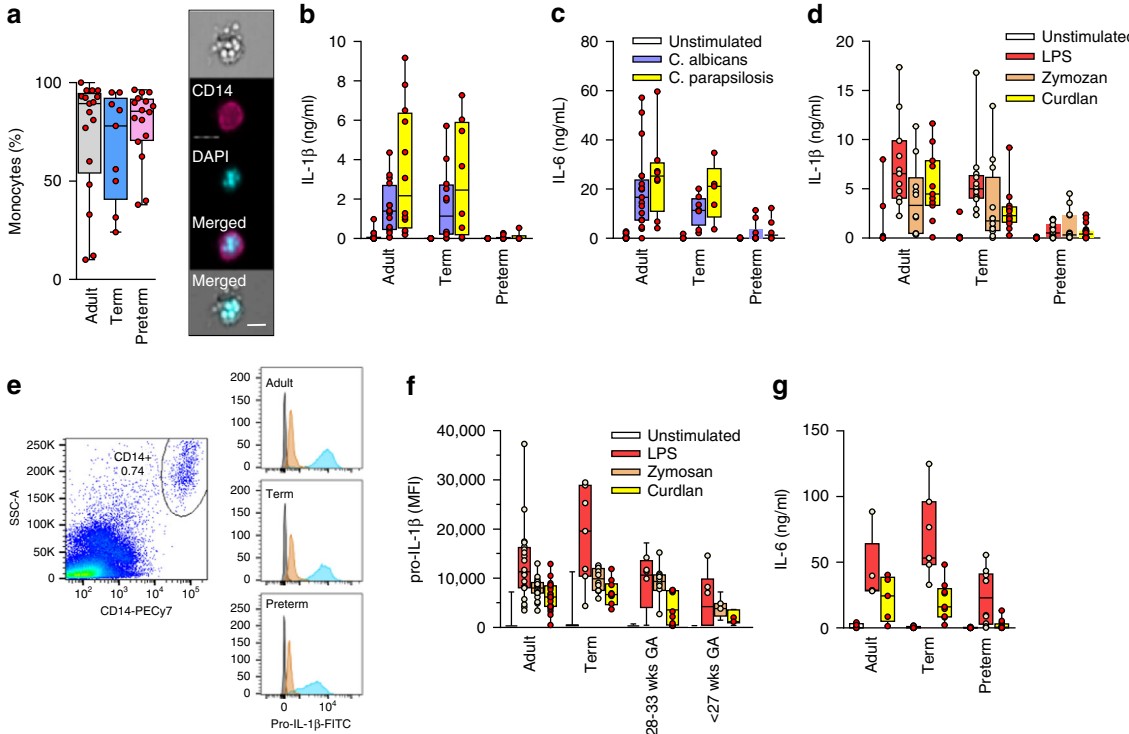

**Fig. 1** Responses to *Candida spp.* in neonatal immune cells. **a** Phagocytosis of Candida in monocytes (boxes and whiskers), including a representative flow microscopy diagram (white bar = ~10 μm). Data pooled from multiple experiments over 14 months (9 to 17 subjects per age group; see Supplemental Data for clinical information on preterm subjects); **b** IL-1β and **c** IL-6 response (blood mononuclear cells) to *C. albicans* or *C. parapsilosis* (24 h stimulation; 10 to 18 subjects per age group; boxes and whiskers); (**d**) IL-1β (24 h; 11 to 21 subjects per age group; boxes and whiskers) and **e** representative gating for pro-IL-1β (5 h LPS stimulation), gated on CD14-expressing cells (black = fluorescent-minus one control; orange = unstimulated; blue = LPS; representative preterm sample is from a 26 weeks' infant); **f** pro-IL-1β (5 h) or **g** IL-6 (24 h) in response to LPS, zymosan or curdlan (mononuclear cells; 11 to 21 subjects per age group; boxes and whiskers); for **b** and **c**, data was pooled from multiple experiments assayed in four ELISA batches with similar distribution of samples per age group

similar kinetics, peaking between 5 and 8 h in all three age groups (Jen R, Sharma A, and Lavoie PM, unpublished data and ref. [24]). Strikingly, however, the strong gene expression response in preterm cells contrasted with the lack of corresponding protein expression (Fig. 4d), particularly for IL-1α/β, TNF-α, and IL-10, suggesting a translation defect. Of note, the same changes in metabolic and ribosomal-related genes persisted in LPS-stimulated cells (Supplementary Data 2).

**Defective immune response translation in preterm monocytes**. To assess whether preterm monocytes lacked translation of immune response genes, we performed polysome analysis. Monocyte lysates from all three age groups were subjected to sucrose gradient fractionation; the concentration of specific mRNAs in monosome, disome, and light and heavy polysome fractions were measured by RT-qPCR. This experiment was extremely difficult due to the limited number of monocytes obtainable from preterm infants. Given the lack of dectin-1 response, we focused on interrogating this pathway (Fig. 5a).

Despite the technical challenge, we were able to perform polysome profiling in all three age groups (Fig. 5b). In general, the overall distribution of mRNAs across the polysome gradient was similar between preterm, term and adult samples, but distinct between genes, suggesting that each mRNA is translated with different intensity (Fig. 5c). Expression of the *Malt1*, *Bcl10*, and *Card9* genes in total mRNA fractions were comparable between all three age groups (Fig. 5d). In the polysome fractions, *Actb*, *Clec7a*, and *Card9* mRNAs, which respectively encode β-actin, the dectin-1 receptor, and Caspase recruitment domain-containing

protein 9 (CARD9), were present in heavy polysomes in preterm monocytes. This is similar to the distribution in term and adult monocytes, indicating that these mRNAs are translated even at low gestational age. In contrast, the majority of *Bcl10* and *Malt1* mRNAs was associated with monosomes and disomes in preterm and adult monocytes, and only *Malt1* mRNA was detected in heavy polysomes, in term monocytes (Fig. 5c). These results potentially indicate that dectin-1 signaling genes are differentially translated across the gestational age spectrum.

Next, we asked whether these proteins were expressed in monocytes from all three age groups. The dectin-1 protein was comparably expressed between preterm, term, and adult monocytes (Fig. 5e). In contrast, expression of the MALT1 protein, but also the Syk and CARD9 proteins, were severely reduced in preterm monocytes (Fig. 5f, g; Supplementary Figure 6). For Bcl10 protein, expression was undetectable in preterm monocytes by Western blotting (Fig. 5f, h). To confirm whether translation was reduced in preterm cells, pulse-labeling experiments were conducted and showed reduced 35S-methionine/cysteine incorporation in the latter, in response to LPS (Supplementary Figure 7).

**Role of MALT1 in the monocyte response against *Candida spp***. The role of the MALT1 signalosome in dectin-1 responses in myeloid cells has been mainly studied in mice macrophages and dendritic cells, but it has not been formally established in human monocytes[20,25]. Moreover, due to the major role of MALT1 in T and B cells activation, it is also unclear whether signaling through

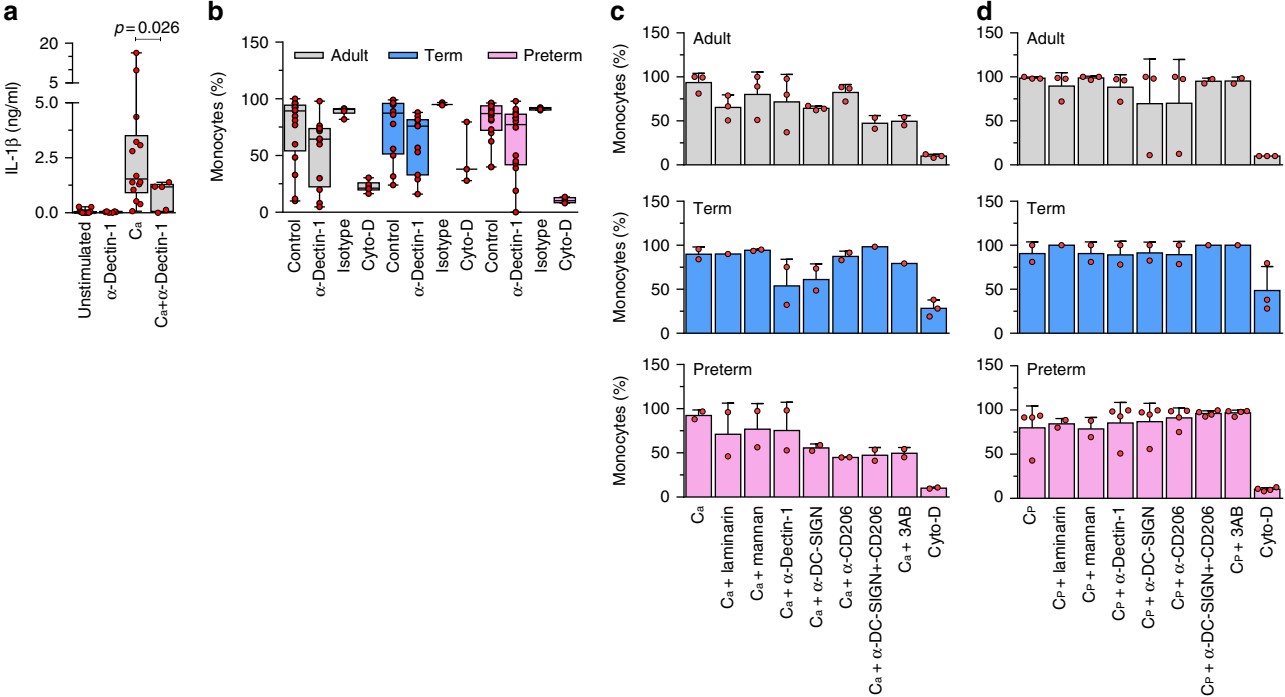

**Fig. 2** Importance of dectin-1 in response to Candida in neonatal monocytes. **a** Antibody blocking of dectin-1 reduces cytokine production to *C. albicans* ($C_a$) in adult mononuclear cells (one-sided *t*-test with Welch's correction for unequal variance; 6 to 12 subjects per condition; boxes and whiskers); **b** Phagocytosis of *C. albicans* upon blocking with anti-dectin-1 receptor antibody, same-isotype control or cytochalasin-D (Cyto-D); (11 to 18 subjects per age group; boxes and whiskers); Blocking of **c** *C. albicans* ($C_a$) or **d** *C. parapsilosis* ($C_p$) phagocytosis using laminarin (blocking dectin-1), mannan (blocking dectin-2 and mannose receptor), anti-DC-SIGN, and anti-CD206 antibodies, or a combination of all three antibodies (bar graph with mean ± standard deviation (SD); 2 to 3 subjects per age group)

this molecule is also essential for myeloid responses to *Candida* in humans[26].

To confirm this, we examined the effect of blocking MALT1 on both cytokine responses and phagocytosis of *Candida species*, in human monocytes. Complete loss of cytokine response to curdlan was observed with MALT1 inhibition (Fig. 6a). When testing responses to clinical strains of *Candida*, MALT1 blocking also completely abrogated cytokine responses to *C. albicans* or *C. parapsilosis* (Fig. 6b–d). Blocking of Syk also partially abrogated responses to these two micro-organisms, indicating that the response to *Candida* primarily involves signaling through dectin-1 (Fig. 6b–d). On the other hand, these responses were neither affected by blocking of MyD88 (downstream of TLRs) nor Raf-1 (mediating MALT-1-independent dectin-1 signaling; Fig. 6b–d).

Given that preterm monocytes are fully able to phagocytose *Candida*, we also examined the role of MALT1 in this function. MALT1 inhibition did not block uptake of *Candida spp.*, indicating a non-essential role in phagocytosis (Fig. 6e). Altogether, these results confirm an essential role of the MALT1 signalosome in the recognition, but not the phagocytosis of whole *Candida* in human monocytes.

**Altered energy metabolism in preterm monocytes.** Recent data has demonstrated that a shift in basic cellular energy metabolism from oxidative phosphorylation to glycolysis is required during immune cell activation, in order to provide the rapid energy and metabolic intermediates necessary for translation of immune response proteins[27]. While these studies have been mainly conducted in macrophages and dendritic cells, we confirmed the importance of this mechanism also in human monocytes by showing inhibition of cytokine production with 2-DG, a non-functional glucose analog that inhibits glycolysis (Fig. 7a).

In light of the reduced translation in preterm monocytes, we asked whether glycolysis could be impaired. To this end, we compared the glycolytic capacity of preterm, term, and adult monocytes. Notably, glycolysis was severely diminished in preterm monocytes (Fig. 7b, c). The reduced glycolytic activity in preterm monocytes was also reflected in reduced lactate production (Fig. 7d) and reduced glucose uptake (Supplementary Figure 8), at lower gestation, upon LPS but also at rest (in unstimulated cells). In term monocytes, glycolysis was variably affected, suggesting a transitional functional state.

Given that phagocytosis is preserved in preterm cells, these data raise an important question: do these metabolic constraints differ between cytokine responses and phagocytosis? Indeed, the requirement for glycolysis and/or translation has been mainly studied for PRR-mediated cytokine responses[27], but only sparsely for phagocytosis[28,29]. Consequently, we showed that phagocytosis of *Candida* was unaffected by blocking glycolysis, or even by blocking of de novo protein synthesis (Fig. 7e). These results indicate a specific requirement of glycolysis for PRR-mediated responses, but not for phagocytosis.

Human adult monocytes are capable of rewiring their metabolism to adapt their energy requirements in a function-specific manner[29]. The nuclear receptor PPAR-γ plays an important role in this function, regulating lipid uptake. PPAR-γ has also been shown to regulate phagocytosis in addition to promoting anti-inflammatory responses in human monocytes[30,31]. In preterm monocytes, we observed a net down-regulation of mitochondrial electron transport (Supplementary Figure 9) and oxidative phosphorylation transcripts (Supplementary Figure 10A). Corresponding measures of oxygen consumption rates in preterm monocytes suggested decreased ATP-linked respiration (Supplementary Figure 10B), consistent with a metabolically quiescent state. Interestingly, expression of PPAR-

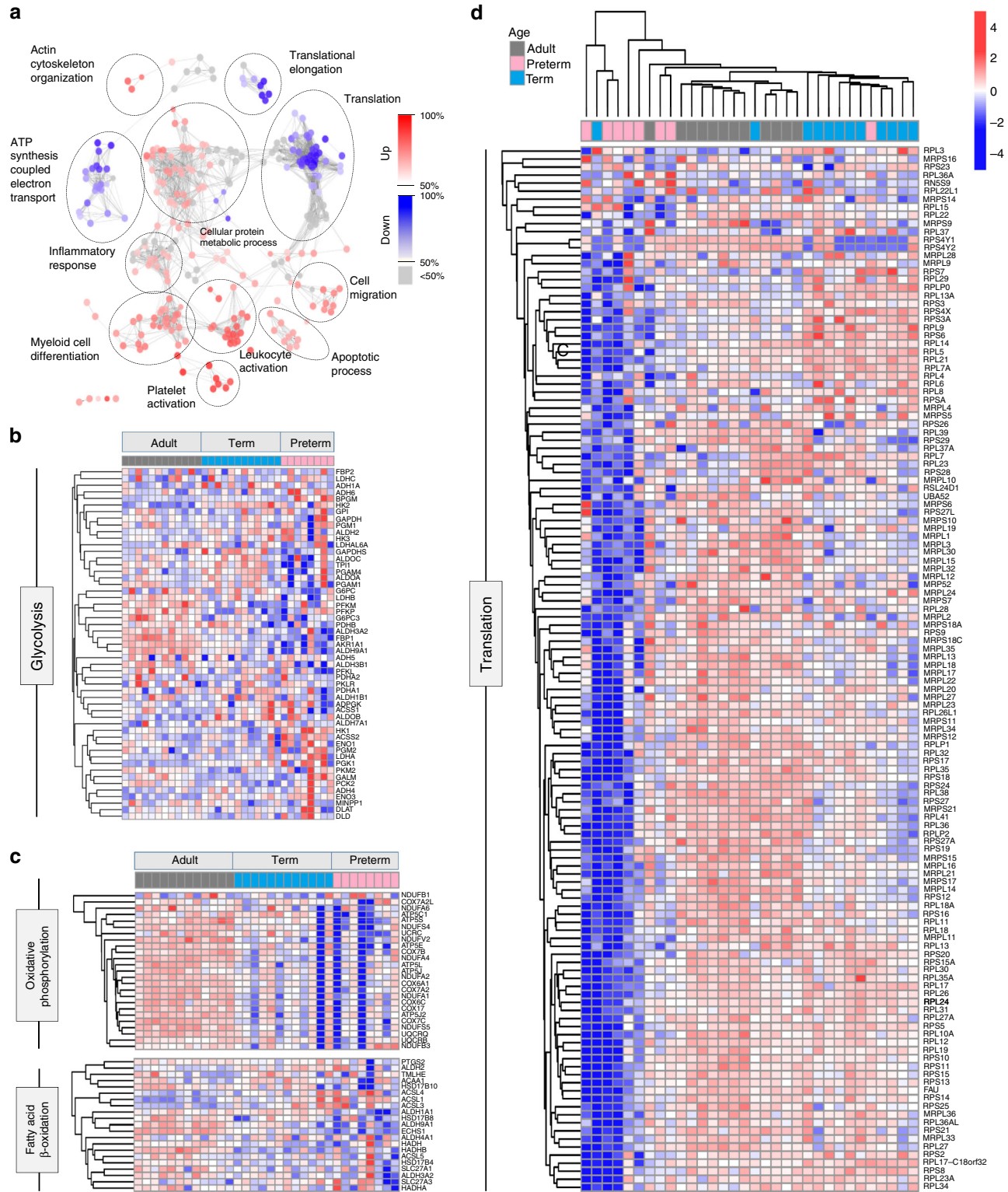

**Fig. 3** Transcriptome analysis in neonatal monocytes. **a** Gene ontology analysis of differentially expressed genes (FDR < 0.01) between unstimulated preterm, and combined term and adult samples (monocytes; $n = 8$ to 12 subjects/age group); Expression heatmap and unsupervised clustering for genes involved in **b** glycolysis, **c** oxidative phosphorylation and fatty acid beta-oxidation and **d** ribosomal proteins. Scale represents $z$-score

γ was increased in preterm monocytes (Supplementary Figure 10A). Furthermore, increased PPAR-γ activity in preterm monocytes is supported also by upregulation of its target genes mapping to metabolic pathways (Supplementary Figure 11). Altogether, these data indicated a rewiring of metabolic pathways favoring phagocytosis instead of pathogen-mediated pro-inflammatory cytokine responses, in preterm monocytes.

**Developmental regulation through mTOR**. The data above raise an additional important question: how are glycolysis and

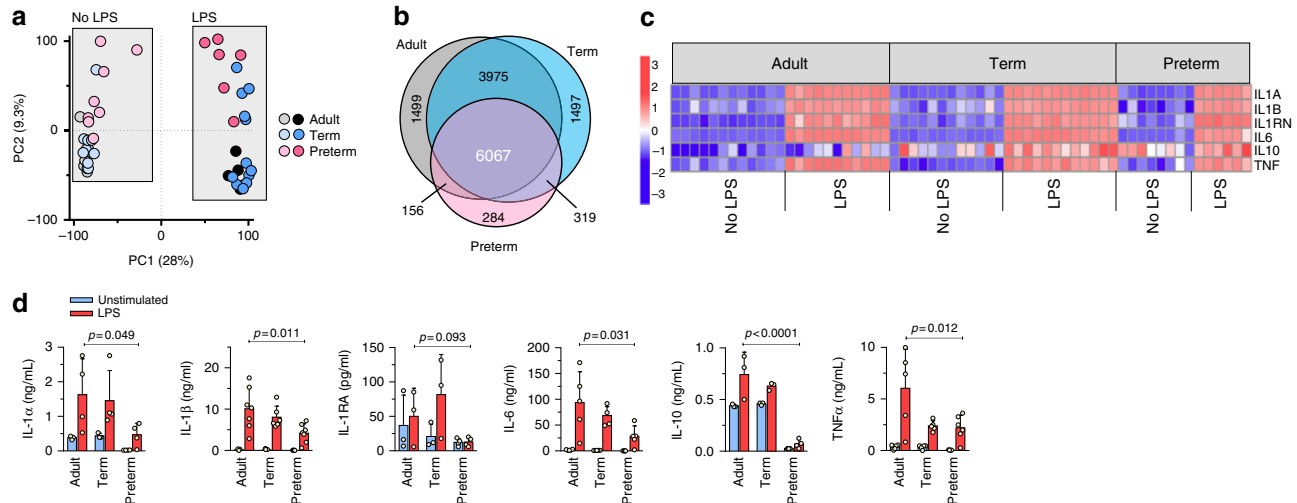

**Fig. 4** Transcriptome and cytokine responses to LPS in neonatal monocytes. **a** Principal component (PC) analysis of unstimulated versus LPS-stimulated monocytes and **b** Venn diagram of differentially expressed genes (LPS-stimulated samples) overlapping between age groups (FDR 5%). **c** Expression heatmap of cytokine genes (5 h LPS; scale = z-score) Data from same subjects as in Fig. 3, except for 1 adult and 2 preterm subjects with insufficient cells for LPS condition; **d** Production of cytokines following LPS stimulation (24 h, mononuclear cells) by ELISA (effect of age by two-way ANOVA; mean ± SD; n = 3 to 6 subjects per age group)

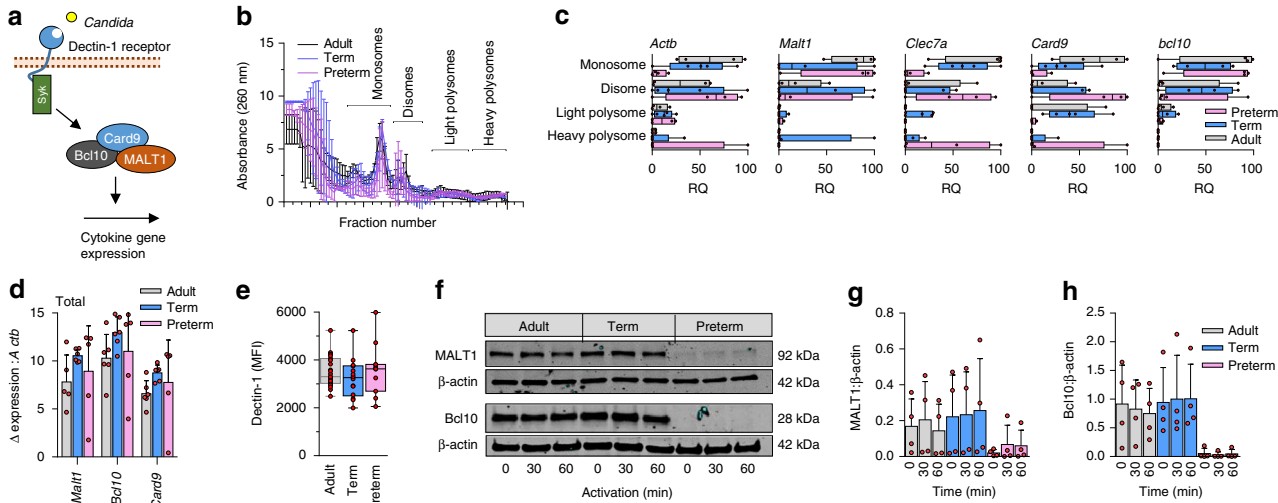

**Fig. 5** Gene expression and translation of dectin-1 signaling proteins. **a** Illustration of selected signaling molecules downstream of dectin-1; **b** Polysome profiles and **c** quantification of signalosome genes (qPCR) in monosome, disome, and light and heavy polysome fractions (monocytes). Data are from 4 subjects per age group (boxes and whiskers; RQ = relative quantification); **d** Quantification of signalosome genes (qPCR) in total RNA fractions (4 to 5 subjects/age group; mean ± SD); **e** Surface expression of dectin-1 (flow cytometry, mononuclear cells, gated on CD14-expressing cells; data pooled from 10 to 23 subjects per age group; boxes and whiskers); **f** Representative (cropped) Western blot of MALT1 and Bcl10 protein expression in monocytes after 0 to 60 min LPS stimulation. Representative blot is from a 29 weeks gestation sample. Images cropped from same blot probes with each antibody; cumulative quantification of 4 independent Western blot experiments for **g** MALT1 and **h** Bcl10 (mean ± SD)

translation negatively regulated in preterm monocytes. Mechanistic target of Rapamycin (mTOR) is a key regulator of the metabolic switch towards glycolysis during immune activation (depicted in Fig. 7f)[32]. Therefore, we examined the mTOR regulator node developmentally. Interestingly, preterm monocytes showed reduced phosphorylation of mTOR following LPS stimulation (Fig. 7g). These cells also displayed reduced expression of its main downstream target 4EBP1, which is important in mediating the effect of mTOR on translation downstream of PRR activation (Fig. 7g). Given the difficulty in getting sufficient amounts of monocytes for efficient detection of mTOR phosphorylation by Western blot in preterm cells, we employed a flow cytometry assay, on a larger number of subjects. Using this approach, we confirmed the reduced mTOR phosphorylation, but

also the gestational age-dependent functional dissociation between mTOR phosphorylation and cytokine responses in preterm monocytes (Supplementary Figure 12).

Next, we more closely examined regulatory gene expression within the mTOR pathway. Again, due to the absence of dectin-1/MALT1 signaling, we focused our analyses on LPS-stimulated cells. Major differences were observed in neonatal monocytes in the mTOR pathway (Supplementary Data 3), including a decreased expression of the upstream mTOR activator RAC-alpha serine/threonine-protein kinase (encoded by the *Akt1* gene). Expression of the insulin receptor substrate 2 (*Irs2*) gene, which is an upstream regulator of mTOR, the genes encoding the adenosine monophosphate-activated protein kinase (*prkaa2*) and the hypoxia-inducible factor 1-alpha (*Hif1a*) were increased in

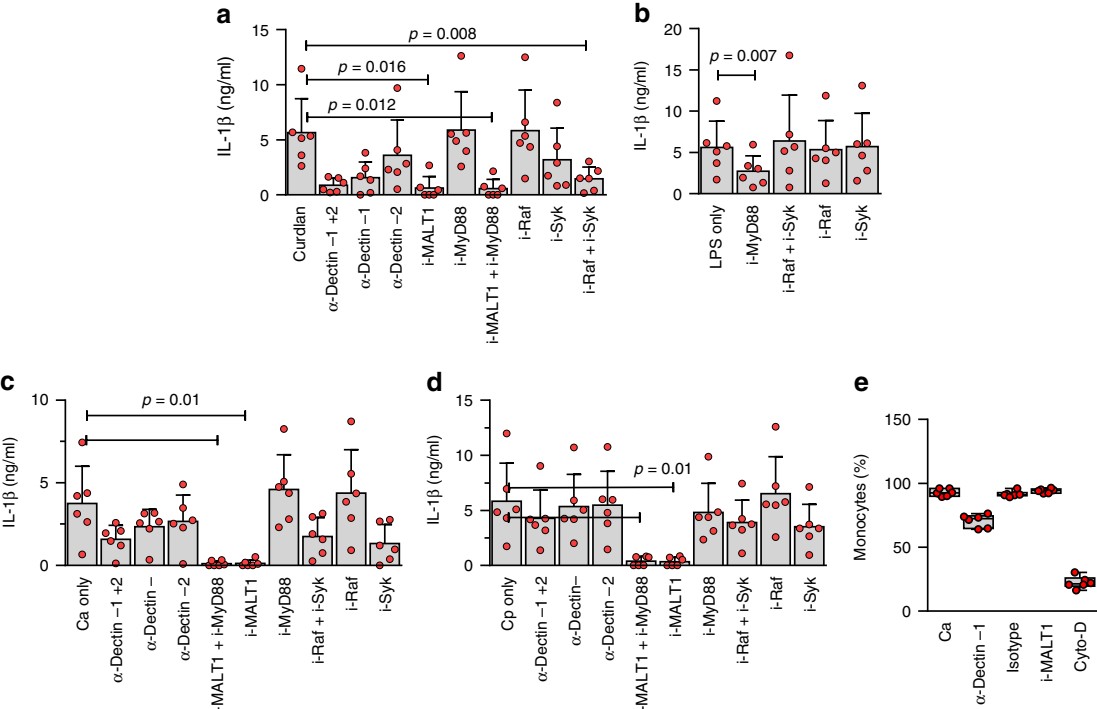

**Fig. 6** MALT1 is essential for *Candida* recognition and curdlan signaling in human monocytes. IL-1β production after stimulation (24 h) with **a** curdlan or **b** LPS, or with **c** *C. albicans* or **d** *C. parapsilosis*, and upon inhibition (i) of MALT1 (using mepazine hydrochloride), MyD88 (ST2825), Raf (GW5074) or Syk (Piceatannol), or blocking of dectin-1 or dectin-2 using antibodies (data pooled from 6 experiments (6 subjects); mean ± SD); **e** Effect of MALT1 inhibition, dectin-1-blocking antibody or actin polymerization inhibition (using cytochalasin-D) on phagocytosis of *Candida* by monocytes (boxes and whiskers; data also from 6 experiments). Statistical significance was calculated using 2-sided paired *t*-tests

preterm monocytes compared to term or adults especially after LPS stimulation, which may represent compensatory mechanisms.

Most notably, expression of the negative mTOR regulators NAD-dependent deacetylase sirtuin-1 (*Sirt1*) and DNA damage inducible transcript-4-like (*Ddit4l*) were profoundly upregulated in preterm monocytes. *Ddit4l* is a paralog of the DNA damage inducible transcript-4 (*Ddit4*) that has also been shown to inhibit mTOR[33,34] (Fig. 7h). Upregulation of *Ddit4l* was further confirmed at the protein level in neonatal preterm monocytes by Western blot (Supplementary Figure 13), suggesting that this molecule may represent an important developmental-specific negative mTOR regulator.

**Glycolysis is required for MALT1 protein expression.** The experiments described above reveal two main deficits in preterm monocytes: (i) a lack of dectin-1 signaling resulting in a lack of cytokine responses to *Candida*, and (ii) a reduced metabolic state leading to reduced innate immune responsiveness to LPS activation. In order to investigate whether these two functional deficits are linked, we tested whether blocking of glycolysis impaired translation of MALT1. In both Western blot and flow cytometry analyses, MALT1 protein levels were significantly reduced after 2-DG treatment (Fig. 8a–c). In contrast, protein expression of β-actin, a highly abundant gene transcript (see above) was unaffected (Fig. 8a). Conversely, inhibition of MALT1 did not affect the glycolytic capacity of these cells, as shown by sustained lactate levels both at rest and after LPS (Fig. 8e). In the latter experiment, LPS-induced cytokine responses correlated with lactate levels (Fig. 8f). Together these data are consistent with reduced glycolysis limiting cytokine responses and MALT1 protein expression in preterm cells rather than the other way around.

**Immune response and risk of invasive *Candida* infection.** Finally, we sought to determine how reduced cytokine responses in preterm cells may impact infants' risk of invasive infections. In order to assess this, we reviewed data from 39,336 infants born below 33 weeks of gestation in Canada over 10 years. We found that rates of invasive infections exponentially increased with decreasing gestational age (Supplementary Table 1). When specifically looking at *Candida* infections, rates of candidemia also increased exponentially at lower gestation (Supplementary Table 1). Rates of invasive *Candida* infections also inversely correlated with in vitro responses to this pathogen (Supplementary Figure 14). Altogether, our data provide further evidence that the increased risk of invasive infections in these infants may be related to a lack of immune responses to *Candida species*.

## Discussion

Infection and prematurity are thought to kill more newborns than any other cause globally (about one million infants die from infections each year)[35]. Despite this, we lack insights into the very basic mechanisms responsible for the high immunological vulnerability of preterm neonates, which restrain innovation towards new therapeutic interventions. In this study, we provide the first evidence of a role for cellular energy metabolism regulating neonatal innate immune responsiveness during ontogeny, in a gestational age-dependent manner. We also provide insights into how this may occur, through a lack of expression of key immune response proteins in the context of reduced glycolytic metabolic capacity. Our data suggest that multiple PRR responses could be broadly affected through these mechanisms in preterm myeloid cells. In light of our data, we posit that constraining innate immune reactivity in utero during development is physiologically important, possibly to limit adaptive immune co-activation signals during the early establishment of self-tolerance, but also to

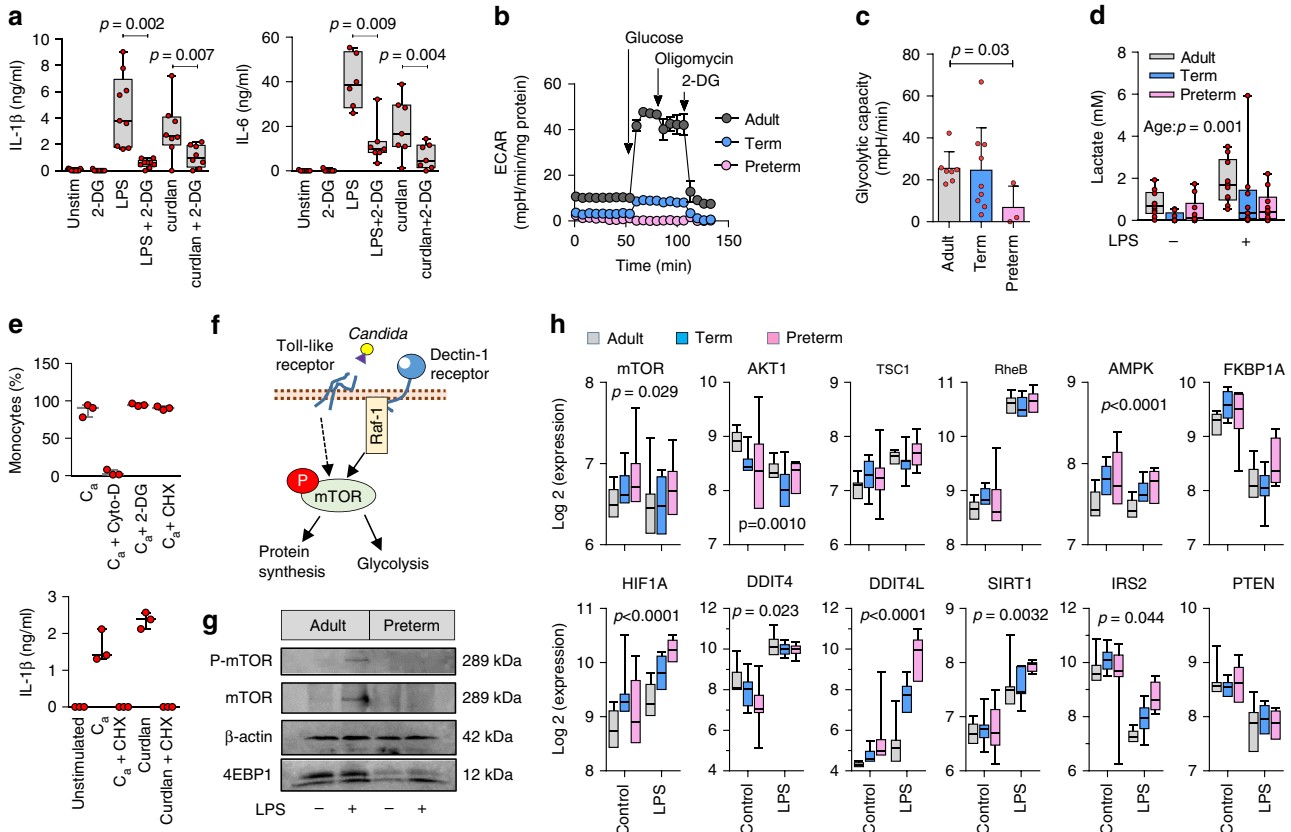

**Fig. 7** Impaired glycolysis and mTOR activity in preterm neonatal monocytes. **a** Effect of blocking glycolysis on cytokine responses to LPS stimulation (24 h) in mononuclear cells (2-sided paired *t*-tests; boxes and whiskers; 4 to 9 samples per condition; 2-DG = 2-deoxy-d-glucose); **b** Extracellular acidification rates (ECAR, normalized to total protein content), under baseline glucose-free conditions, after addition of glucose, oligomycin, and 2-DG (representative experiment from a 29 week gestation preterm sample); **c** Glycolytic capacity (cumulative data pooled from 4 independent experiments; 3–9 subjects per age group; mean ± SD; p value by Mann–Whitney *U* test); **d** Lactate production in mononuclear cells after LPS stimulation (24 h); *p* value = effect of age by 2-way ANOVA. Data from 9 to 10 subjects per age group; boxes and whiskers; **e** Effect of blocking glycolysis or translation (using cycloheximide, CHX) on the phagocytosis of *C. albicans* (C$_a$); boxes and whiskers; 3 subjects; Cyto-D = cytochalasin-D; **f** Depiction of signaling events between Toll-like receptor and Raf-1-mediated dectin-1 activation, mTOR phosphorylation, and increased glycolysis and protein synthesis; **g** Western blot image (cropped) of mTOR/4EBP1 expression and mTOR phosphorylation (monocytes; preterm sample from 29 weeks gestation; cropped images from same blot probed for mTOR, β-actin, and 4EBP1, and stripped/re-probed for phospho-mTOR); **h** Expression of mTOR-related genes (monocytes; 6–12 per age group; boxes and whiskers; 2-way ANOVA across age groups, only significant *p* values are shown)

avoid energetically costly immune responses early in gestation. Indeed, translation consumes up to ~45% of cellular ATP[36]. Immune activation triggers an upregulation of glycolytic pathways to rapidly supply the high energy needs and metabolic intermediates required for protein synthesis[37]. Preventing an invasive *Candida* infection is also relatively costly energetically[28], and may not be specifically advantageous prior to a viable fetal stage. On the other hand, the preservation of other immune functions such as phagocytosis may be more essential and is in keeping with previous studies[29,38,39]. This may stem from a more essential need for house-keeping tissue remodeling functions during fetal development[40].

Our results raise important mechanistic questions. For instance, it is unclear how translation is selectively impaired by the metabolic constraints observed in preterm monocytes. Intriguingly, transcripts that were less abundant (by qPCR) also showed less protein expression, consistent with a limited ribosomal activity in preterm cells. We hypothesize that the reduced ribosomal activity in preterm cells may selectively impair the translation of immune response genes that are more scarcely abundant, as suggested in our polysome experiments. On the other hand, abundant transcript may still be efficiently translated, allowing to maintain cell viability. Upon activation of immune

cells, it is also possible that immune response genes are selectively not translated through sequence-specific mechanisms. Supporting this model, 5'-terminal oligopyrimidine tract (TOP)-containing mRNAs such as those encoding ribosomal proteins and translation factors, and mRNAs with short 5'UTRs have been shown to be more sensitive to mTOR activity[41–43]. In such cases the availability of eIF4F complexes (via regulating 4E-BP) may specifically limit the synthesis of those proteins[41–43]. In preterm cells, it is also possible that under reduced mTOR activity, the rate of translation of immune response proteins may be less than the rate of their turn-over due to cis-acting elements in their 5'UTRs. Consistent with this model, pharmacological inhibition of mTOR using PP242, reduced translation of MALT1[43]. The precise mechanisms that limit expression of immune proteins in preterm cells require future studies.

Our data shed additional light on the roles of MALT1 in the response of myeloid cells to *Candida spp.* in humans. Previous studies have been conducted in murine dendritic cells where blocking of MALT1 using siRNA or using a knock-out in mice abrogates cytokine responses to multiple strains of *Candida spp.* as well as to curdlan[20,25]. In humans, the role for MALT1 in protecting against invasive *Candida* infections has been obscured by the major importance of MALT1 in T cells and B cells[26]. Our

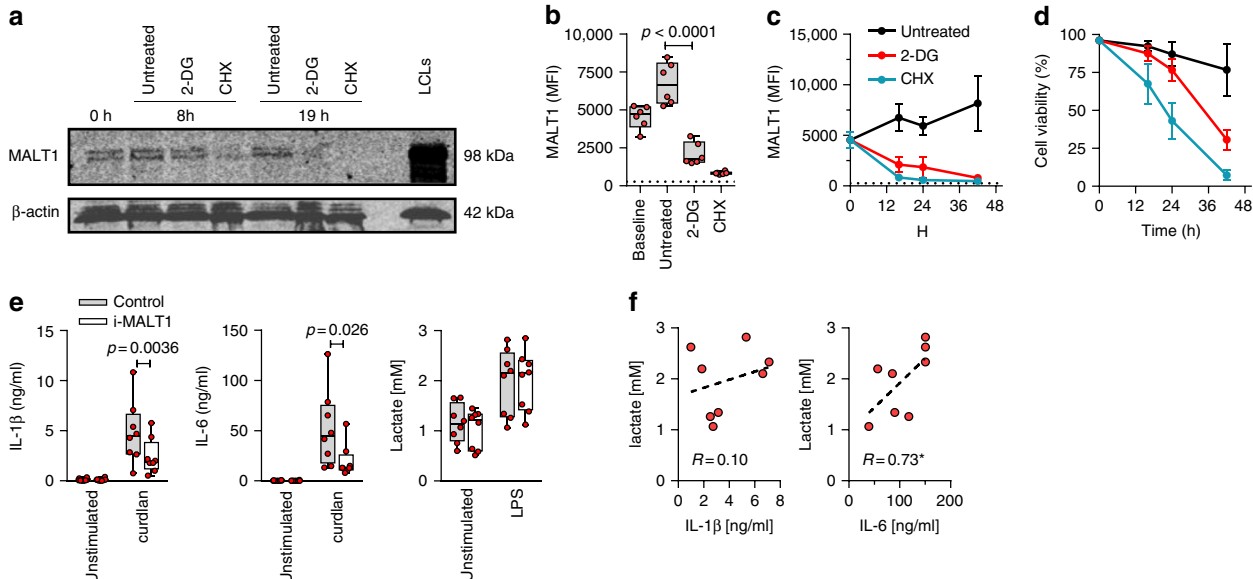

**Fig. 8** Inhibition glycolysis results in loss of MALT1 protein expression. Effect of blocking glycolysis (using 2-DG) or of blocking translation (using cycloheximide, as control) on MALT1 protein expression (monocytes). **a** MALT1 protein was detected by Western blot (left panel; representative of two experiments; cropped images from same blot probed with each antibody) at 8 h and 19 h. Lymphoblastoid cell line (LCL) lysate used as positive control for MALT1 protein expression; MALT1 protein detection (**b**) at 16 h and (**c**) over time (intracellular staining by flow cytometry, gated on CD14-expressing cells; MFI mean fluorescence intensity; dotted line: signal for fluorescence-minus-one staining control MFI level; boxes and whiskers with a paired 2-sided $t$-test in **b**; mean ± SD in **c** and **d**; **d** corresponding cell viability over time (mean ± SD); 6 subjects. **e** Effect of MALT1 inhibition on IL-1β, IL-6, and lactate production at rest and following LPS (mononuclear cells; boxes and whiskers with 2-sided paired $t$-tests); **f** correlation between LPS-induced IL-1β and IL-6, and lactate production (Spearman' $r$; *$p < 0.05$; with dotted regression line); 8 subjects. All experiments were conducted in adult cells

work confirms a central role for MALT1 in the immune recognition of *Candida* also in human myeloid cells. Moreover, our data suggest that the loss of MALT1 function in preterm cells impairs the ability to respond to *Candida spp.*, although the extent to which impaired translation of additional immune response genes limit other immune pathways remains to be determined. Of note, our observation that *Candida* responses were partially abrogated by blocking Syk may suggest the existence of a Syk-independent, yet unidentified, non-dectin receptor that activates responses via MALT1.

Our data have potential clinical relevance. While a mechanism that limits innate immune responsiveness during fetal development may be essential to protect against organ damage in utero[44], it becomes a serious problem when a preterm birth occurs, due to increased infections in the premature infant. One key question is: can these constraints be reversed therapeutically, and if so, how? The comparably strong transcriptomic response in preterm monocytes is not unique to LPS, and has been reported with whole bacterial micro-organisms (*Escherichia coli* and *Staphylococcus epidermidis*)[45]. Monocytes from adult subjects after an endotoxin challenge or sepsis also display a similar transcriptome profile[46]. In this latter case, immune training using low dose stimulation through the dectin-1 receptor could reverse the metabolic constraints[46], though the absence of dectin-1 signaling in preterm cells likely precludes this approach[47]. Alternatively, metabolic constraints may also be reversed with pre-treatment of cells with IFN-γ[46,48–50]. Interestingly, clinical trials suggest that IFN-γ treatment may ameliorate clinical outcomes in adults post sepsis[51,52]. Newborn immune cells produce reduced IFN-γ, especially in absence of exogenous polarization[53]. One study showed that preterm whole blood pre-treated with IFN-γ show augmented immune responsiveness to LPS in vitro, thus representing a potential therapeutic avenue that could be explored to restore immune defenses after birth[54]. Alternatively, SIRT1

inhibitors have also been shown in pre-clinical trials to augment glycolysis and immune responses post sepsis, through a stabilization of HIF-1α[55]. Despite the multiple challenges that lie between our studies and a potential therapeutic application in preterm infants, our studies represent an important discovery and reveal potential translational research avenues to tackle the high vulnerability of these infants to infections. However contrary to adults, our data suggest that the lack of glycolytic and mTOR activity in preterm cells may be actively suppressed, possibly through the putative mTOR regulator DDIT4L. In this context, further studies are required to determine how a metabolic reprogramming of preterm myeloid cells can be sustainably achieved at the cellular level.

A limitation of our study is the strict use of cord blood which may not adequately represent peripheral blood immune responses during the neonatal period. However, obtaining sufficient volumes of peripheral blood from premature neonates is ethical questionable and could not be achieved for safety reasons. Indeed, only a handful of studies have measured immune parameters in peripheral blood from premature neonates, and these studies confirm the marked attenuation in PRR responses compared to term neonates, with correlation of the expression of phagocytosis and monocyte activation marker between cord and peripheral blood[38,56–60]. Therefore, it appears that cord blood is a reasonable alternative to using peripheral blood, though important differences exist and the former does not reflect the dynamic changes occurring over time when infants are most susceptible to sepsis[56,60]. A second limitation is the focus on monocytes, which incompletely recapitulates the complex cell-microbes interactions during sepsis in vivo. For example, IL-1β can be cleaved by serine proteases (e.g., PR3, cathepsin G) expressed by neutrophils, in addition to caspases. Because of the crucial role of neutrophils in preventing *Candida* invasion in humans[8], specific studies are required to determine the contributions of other immune cells.

In summary, our data reveal important metabolic constraints that limit immune activation against major neonatal pathogens in monocytes before the term of gestation. This mechanism provides a unifying pathway to explain the broadly suppressed innate immune responsiveness during human fetal development. Future studies are required to understand how reduced mTOR function impairs translation in preterm monocytes, and how the putative negative regulator of mTOR, DDIT4L may regulate this process[34]. Finally, the availability of metabolic reprogramming drugs in pre-clinical studies may offer therapeutic avenues to augment immune reactivity and reduce the major burden from infections in these high-risk preterm newborns[55].

## Methods

**Human subjects**. This study was conducted according to the Canadian Tri-Council Policy Statement: *Ethical Conduct for Research Involving Humans* at http://www.pre.ethics.gc.ca/eng/policy-politique/initiatives/tcps2-eptc2/Default/. Cord blood was obtained from preterm infants (<33 weeks) and term infants (>38 to<41 weeks) born by elective cesarean section in absence of labor, and peripheral blood was obtained from healthy adult volunteers recruited at the BC Children's Hospital Research Institute. All blood samples were collected in sodium heparin vacutainers. Cord blood from more than 104 preterm deliveries were used in the experiments included here. To assess whether the lack of dectin-1 response in preterm subjects may have been due to an enrichment of loss-of-function genetic variants in the dectin-1 receptor, genotyping for the common Y238X mutation (SNP rs16910526) was performed on a previous cohort of infants born before 31 weeks of gestation[61]. Only 2 out of the 177 infants were homozygous for the rare variant, a rate similar to the adult population[62].

The clinical characteristics of the preterm subjects included in each of the figures are provided in aggregated form, in the accompanying Supplementary Tables 2 to 14. All cord blood samples were collected following delayed cord clamping clinical standards, and after the placenta was detached from the infant. Written consent was obtained from all participants, except for eight preterm cases where parents were unavailable/non-reachable in person or by phone. In these cases, no clinical data was obtained on completely de-identified cord blood samples, in accordance with the University of British Columbia Children's & Women's Research Ethics Board (C&W REB). The study complied with all relevant ethical regulations and was approved by the C&W REB (#H07-01698).

**Cells**. Mononuclear cells (MNCs) were isolated from between 0.5 and 20 mL of whole blood (depending on each ensuing experiment) by density centrifugation using Lymphoprep™ (STEMCELL Technologies). Monocytes were purified from MNCs generally from at least 1 mL of whole blood, using an EasySep™ positive CD14 selection kit (STEMCELL Technologies, #18058) and 1 mM EDTA to prevent cell clumping. Monocyte purity was regularly determined by flow cytometry (FSC/SSC plus staining for CD14 surface expression) throughout the study, and strictly confirmed to be > 95% for example, for microarray studies.

**Reagents**. Pediatric clinical isolates of *Candida albicans* and *Candida parapsilosis*, identified by mass spectrometry (Bruker Daltonics, Billerica, MA) were obtained from the BC Children's Hospital Microbiology Lab (Vancouver, Canada). We used fixed *Candida spp.* yeast particles which are closer antigenically to yeast forms[5]. Fungi were grown over 4 days at 30 °C in BHI (Brain-Heart Infusion) broth (Oxoid, Nepean, ON). Yeast in exponential growth phase were harvested after 4 days, centrifuged and fixed in 10% paraformaldehyde. To ensure reproducibility, batches of *Candida* particles were prepared and used throughout the study. Human dectin-1 and dectin-2 neutralizing antibodies were purchased from R&D Systems (#MAB1859) and Invivogen (#Mabg-hdect2), respectively, and used at 5 µg ml⁻¹ for cytokine studies, and 10 µg ml⁻¹ for phagocytosis assays. Human CD-206 neutralizing antibody was purchased from Biolegend (#321111) and used at 10 µg ml⁻¹ for phagocytosis assays. Human DC-SIGN neutralizing antibody was purchased from Invivogen (#mab-hdcsg) and used at 10 µg ml⁻¹ for phagocytosis assays. Curdlan (β-1, 3-glucan from *Alcaligenes faecalis*), a dectin-1-specific agonist, was obtained from Wako (#032-09902) and used at 10 µg ml⁻¹ in cytokine studies. Lipopolysaccharide (LPS), a TLR-4 agonist purified from *Escherichia coli*, was obtained from InvivoGen (#tlrl-eblps) and used at 10 ng ml⁻¹. Zymosan, a TLR-2/dectin-1 agonist was obtained from Invivogen (#tlrl-zyn) and used at 10 µg ml⁻¹. Mannan was purchased from Sigma-Aldrich (#M3640-1G), and used at 3 mg ml⁻¹ in phagocytosis assays. Laminarin was purchased from Sigma-Aldrich (#L9634-500mg), and used at 5 µg ml⁻¹ in phagocytosis assays. The following anti-target antibodies were used for Western blots (source): Syk (Abcam, #Ab155187), MALT1 (Cell Signaling Tech., #2494 S), BCL10 (Cell Signaling Tech., #4237 S), CARD9 (Abcam, #Ab133560), phospho-mTOR (Cell Signaling Tech., #5536 P), mTOR (Cell Signaling Tech., #4517 S), 4EBP1 (Cell Signaling Tech., #9644 P), β-actin (Abcam, #Ab75186), goat anti-mouse (LI-COR Odyssey, #926-68070) and anti-rabbit antibodies (#926-32213). Primary antibodies were used at a 1:1000 dilution and secondary antibodies were used at a 1:10000 dilution. For MALT1

protein detection by flow cytometry, we used a Phycoerythrin-labeled anti-MALT1 antibody at a 1:100 dilution [EP603Y] (Abcam, #Ab210982). Cell viability was measured using a Fixable Viability Dye eFluor™ 780 (eBioscience; #65-0865-14; 1:100 dilution). An example of gating strategy for MALT1 detection by flow cytometry is provided in Supplementary Figure 15.

**Cytokine stimulation**. MNCs or monocytes were cultured in RPMI 1640 (Roswell Park Memorial Institute medium) supplemented with 2 mM L-glutamine (Gibco, #25030), 10% human AB serum (Gibco #11875-093) and Pen-strep (Gibco, #15140-122) (referred to as complete RPMI medium, or cRPMI). MNCs were stimulated in round-bottomed 96-well plate (Corning Life Sciences, #3799) with fixed *Candida* (multiplicity of infection, MOI = 5), LPS (10 ng ml⁻¹), Curdlan (10 µg ml⁻¹) or Zymosan (10 µg ml⁻¹) for 24 h unless otherwise specified. For enzyme-linked immunosorbent assays, supernatants stored at -80 °C were analyzed in batches for IL-1β and IL-6 (eBioscience, #88-7261-76), or using a multiplex ELISA assay for IL-1α, IL-1β, IL-1RA, IL-6, TNF-α, and IL-10 (R&D Systems) as per manufacturer's instructions. For blocking experiments, MNCs were pre-incubated for 1 h at 37 °C with neutralizing anti-dectin-1 or anti-dectin-2 antibodies, mepazine hydrochloride (10 µM, EMD Millipore #5005000001), cycloheximide (100 µg ml⁻¹, Sigma-Aldrich #C4859), ST 2825 (a MyD88 inhibitor, 10 µM, AdooQ Bioscience, #A15248-1), GW5074 (a Raf-1 inhibitor, 1 µM, Cayman Chemical #10010368-1), Piceatannol (a Syk inhibitor, 20 µM, Cayman Chemical, #10009355-5).

**Flow cytometry**. MNCs were washed in PBS (3×) and stained for surface CD14 (eBioscience #25-0149; 1:100 dilution), and dectin-1 (AbD Serotec, #MCA4661A488; 1:100 dilution) as indicated. Intracellular cytokine staining for pro-IL-1β was performed by fixing/permeabilizing cells with Foxp3 Staining Buffer (eBioscience, #005523-00), and staining with an anti-IL-1β antibody (BioLegend, #508208, 1:50 dilution). After 30 min of incubation, cells were washed 3 times in PBS before analyses. Native mTOR expression and phosphorylation was performed on cryopreserved mononuclear cells, in batches in order to reduce variability. In brief, MNCs were thawed, washed in cRPMI (3×) and rested for 1 h. An equal number of cells was stimulated with LPS for 30 min. CD14 antibody (eBioscience, #48-0149-42, 1:100 dilution) was added during the last 15 min of incubation. Incubation was stopped and cells were stained intracellularly using Transcription Factor Phospho Buffer Set (BD Biosciences, #565575) for native (Cell Signaling Technology, #5043 S, 1:50 dilution) and phospho-mTOR (Ser2448, Thermo Fisher Scientific, #501123458, 1:50 dilution). Cells were washed 3 times in PBS before acquisition on a BD LSRII flow cytometer (Becton Dickinson). Live cells were gated on singlets. Data were analyzed using FlowJo v10 (FlowJo, LLC, Ashling OR). CD14-expressing monocytes were identified from live cells based on FMO. Cells were analyzed on a BD Fortessa or a 4-laser customized LSRII flow cytometer (Becton Dickinson). Data were analyzed using FlowJo v10 (FlowJo, LLC, Ashling OR).

**Phagocytosis assay**. MNCs were incubated in 96-well plates in the presence of *Candida* strains pre-stained for 15 min with DAPI (Biolegend, #422801) at room temperature. After a 1-h incubation with *Candida* species, cells were quenched in Trypan Blue/PBS of any residual fluorescence of non-phagocytized *Candida spp*, and then washed once in PBS before staining with a CD14 PE-Cy7-conjugated antibody (eBioscience, #25-0149, 1:100 dilution). Cells treated with cytochalasin-D (10 µg ml⁻¹, Sigma-Aldrich) were used as negative controls. Samples were analyzed on an Amnis Image Stream Imager. Results are presented as the percentage of monocytes (CD14-expressing cells) that have phagocytosed at least one *Candida* particle. Data was analyzed using IDEAS software (v.5.0.343.0)

**Western blots**. After stimulation (LPS or curdlan), monocytes were washed, and then lysed in RIPA buffer in presence of phosphatase/protease inhibitors (Santa Cruz Biotechnology, #sc-24948). Protein quantification was performed using a Pierce 660 nm protein assay (Thermo Fisher Scientific). Lysates boiled in 4X Laemmli buffer (Bio-Rad) supplemented with 2-mercaptoethanol were ran on 4–20% mini-PROTEAN TGX gradient gels (Bio-Rad), transferred to PVDF membranes, and incubated with primary and secondary antibodies. Blots were imaged on a LI-COR Odyssey 9120. Images were acquired using Studio ver. 5.2 and adjusted for brightness. Uncut Western blot images are provided in Supplementary Figures 16 to 19.

**Polysome profiling**. For polysome profiling experiments, a minimum of 3 × 10⁶ monocytes (corresponding to at least 5 mL of cord blood) were pre-cultured with cycloheximide (100 µg ml⁻¹) for 10 min at 37 °C, washed once in PBS/cyclohex-imide and lysed in polysome lysis buffer (Mammalian ARTseq™ Ribosome Profiling Kit, Illumina, #RPHMR12126). Lysates were layered onto 10 to 50% sucrose gradients in buffer (0.3 M NaCl, 15 mM Tris-Cl pH 7.5, 15 mM MgCl2, 0.1 mg ml⁻¹ cycloheximide) in the presence of heparin (1 mg ml⁻¹) and spun in an ultra-centrifuge (SW41, Beckman Coulter) for 3.5 h at 4 °C (35,000 rpm) in order to separate RNA strands based upon ribosomal occupancy. After fractionation, the purified, in vitro transcribed dsRED RNA was proportionally spiked into pooled monosome, disome, or heavy or light polysome fractions as internal control. The primer sequences used are presented in Supplementary Table 15.

**Real-time qPCR**. Total RNA was isolated using TRIzol LS (Thermo Fisher Scientific) followed by chloroform extraction and cleaned using RNeasy Mini spin columns (Qiagen) followed by ethanol precipitation. mRNA was reverse transcribed using SuperScript® III First-Strand Synthesis SuperMix (Thermo Fisher Scientific, #18080400), and qPCR experiments were carried out in triplicates on a ViiA 7 system (Applied Biosystems) using both Power SYBR® Green (Thermo Fisher Scientific, #4368706) or TaqMan. Data were normalized on dsRED ($\Delta$Ct method) as detailed in[63]. Briefly, $\Delta$Ct was calculated for each fraction against dsRED. $2^{-\Delta Ct}$ was then calculated, and the sum of all fractions determined. Gene expression (mRNA) was expressed as ($2^{-\Delta Ct} \times 100$)/sum of each fraction.

**Pulse-labeling experiments**. Monocytes were stimulated for 2.5 h and pulsed with $^{35}$S-methionine/cysteine for the last 30 min of the experiment. Cells were washed with PBS and lysed in RIPA (Santa Cruz Biotechnology); equivalent amounts of protein were loaded on polyacrylamide gel. Blots were imaged on film.

**Metabolic assays**. Glucose uptake in monocytes was measured by flow cytometry. Cryopreserved MNCs were rested for 1 h at 37 °C in cRPMI after gentle thawing, and then counted before stimulation for 2 h and 40 min in presence of LPS (10 ng ml$^{-1}$), with 2-(N-[7-Nitrobenz-2-oxa-1,3-diazol-4-yl]Amino)-2-Deoxyglucose (2-NBDG, 15 μM) added for the last 40 min. Cells were stained with CD14 (eBioscience #25-0149; 1:100 dilution) and CD16 (eBioscience #48-0168; 1:100 dilution) at the beginning of LPS stimulation. Cells were washed twice before acquisition on a LSRFortessa™ flow cytometer (Becton Dickinson). Data was analyzed using FlowJo® (FlowJo, LLC). For extracellular flux analyses, monocytes were plated on Cell-Tak™ (Corning, #354240)-coated XF96 Cell Culture Microplates. Cells were rested in a non-CO$_2$ incubator, and glycolysis stress test was performed as per manufacturer protocol (Agilent Seahorse, #103020-100) using the Seahorse XF$^e$96 Analyzer. Extracellular acidification rate (ECAR) was normalized to protein content. L-lactate was measured by colorimetry (Abcam, #ab65331).

**Microarray analyses**. Total RNA from purified CD14$^+$ monocytes was extracted using the MagJET RNA purification kit (Thermo Fisher Scientific). RNA was quantified using NanoDrop™ 1000 spectrophotometer, and its integrity was determined using the Agilent 2100 Bioanalyzer. Samples were hybridized onto an Illumina HumanHT-12 v4 Expression BeadChip and scanned with an Illumina iScan System (Illumina, San Diego, CA, USA). Data were analyzed using R. Data was preprocessed by quantile normalization and log2 transformation. Probes that were non-expressed (based on their detection $p$-values) were removed. For differential gene analyses, linear models were fit using limma[64]. Probes with FDR ≤ 0.05 were considered differentially expressed.

For ClueGO pathway analyses, genes that were differentially expressed by twofold or more when compared to adult or term samples were included. $p$-values were adjusted for multiple comparisons using the Bonferonni step-down method. GO biological-processes ontology was used. Degree of functional enrichment was determined by sorting enriched terms based on a $p$-value threshold of <0.001. We used GO tree intervals between levels 3 and 8 and Kappa Score Threshold of 0.04. Gene Ontology analysis and Network images were generated using the ClueGO 2.3.3 plugin[65] for Cytoscape 3.5.1[66]. Heatmaps for metabolic pathways were generated based on KEGG pathways.

**Statistical analysis**. For each experiment, sample size was estimated based on the variance from 3 to 5 replicates in any age group (assuming comparable variance between age groups). To avoid a selection bias, we balanced distribution of adult, term, and preterm samples in batch experiments over time. GraphPad Prism v6.07 was used for graphs and statistical analyses. To simplify the figures, we present $p$ values only when indicated by the data. Differences between groups were analyzed using 2-tailed $t$-tests, ANOVA (for groups) or as otherwise specified. Normality of data was assumed in most statistical testing, unless grossly skewed. Statistical significance was considered at $p$ values of <0.05. When data is shown in boxes and whiskers, boxes are spanning from the first quartile to the third quartile, with the center line depicting the median. Whiskers are extending to the highest and lowest points.

## Data availability
Microarray data used in Figs. 3, 4a–c, 7h, Supplementary Data 1, 2 and 3, and Supplementary Figures 3, 4, 5, 9 10A and 11 have been deposited in GEO under accession code GSE104510. All other data, including raw data used in each figure will be provided upon reasonable request to the corresponding author, and provided that the nature of the request complies with our institutional ethics board policy.

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

## Acknowledgements

We thank all the participants who provided data and samples for this study, Mihoko Ladd and Ashton Ellis from the BC Children's Hospital (BCCH) Biobank for participant enrollment, Fudan Miao for the microarray library and data generation, and Drs. Rafick Sekaly, Laura Sly, Bruce Vallance and Stuart Turvey for thoughtful comments. This research was funded by a Canadian Institutes of Health Research grant (MOP-123478 to P.M.L.). P.M.L. is supported by an Investigator Award from the BCCH Research Institute and a Career Investigator Award from the Michael Smith Foundation for Health Research (MSFHR). C.M. is funded by a graduate scholarship from the BCCH Research Institute. M.F.C. received a University of British Columbia Faculty of Medicine Summer Student Research Program Award. Data on the prevalence of fungal infections in Canada was provided by the Canadian Neonatal Network.

## Author contributions

B.K. and C.M. led the majority of the experiments as part of their PhD theses. B.K. drafted the first version of the manuscript with help from C.M. H.F. helped with the *Candida* experiments. H.A. helped with polysome profiling and pulse-labeling experiments. K.L. enrolled research participants and helped with the generation of transcriptome data, together with B.K. E.A.M. and M.F.C. helped with sample processing and data acquisition. E.A.B. helped with experimental design and polysome profiling. M.A.S. provided technical and experimental support for metabolic experiments. P.T. provided clinical strains of *Candida* as well as design input and clinical relevance. R.G.M. helped design polysome experiments and supervised EAB. C.J.R. supervised the generation of microarray data. D.S.L. helped design and supervise metabolic studies. E.J. helped design and supervise the translation experiments, including final polysome profiling and pulse-labeling experiment, and data analysis. P.M.L. developed the original concept of the study together with B.K. and C.M., provided general oversight including data analysis and manuscript writing. All authors have reviewed, edited, and approved the final version of the manuscript.

## Additional information

**Competing interests:** The authors declare no competing interests.

