## [Peer Review File · Nature Communications]

Reviewers' comments:

Reviewer #1 (Remarks to the Author):

The present manuscript investigates the mechanisms that are responsible for the defective antimicrobial responses in premature infants. The focus is the invasive infection with *Candida albicans*, the most common human fungal pathogen that causes severe infections in premature newborn. The subject of the study is very relevant, the approach innovative, and the experiments are well performed. The authors present important new information regarding a role for cellular metabolism and mTOR. However, there are also several questions remaining.

Comments:

1. The authors use as last step of monocyte purification a positive selection using CD14 selection kit. The interaction of monocytes with the CD14 antibody can activate cells. Can the authors confirm the most important metabolic findings using a negative selection kit?
2. The authors report diminished glycolysis in monocytes of pre-term infants. It would be important to report however the behaviour of other important pathways such as oxidative phosphorylation and beta-oxidation. The authors have using Seahorse technology, why do they not report also oxygen consumption rate?
3. Differential roles of glycolysis and oxidative phosphorylation for cytokine production and phagocytosis has been recently reported for adult monocytes (Lachmandas et al, Nature Microbiol 2016). How do pre-term monocytes behave in that respect?
4. The authors briefly present the differential recognition of yeasts and hyphae from *Candida albicans* by PRRs. The recognition of hyphae is also done by TLR2 and TLR4, as described by the authors, but these are not the main receptors. The main recognition receptors for hyphae are still of the C-type lectin receptor family, with important roles for dectin-1 and dectin-2.
5. The processing and activation of IL-1 β is actually more complex than described by the authors. In addition of cysteine proteases (such as caspase-1 and caspase-8), a major pathway of activation is through cleavage by serine proteases (e.g. PR3, cathepsin G, etc) especially from neutrophils. As neutrophils are crucial cells for anti-*Candida* host defense, this inflammasome-independent pathway of IL-1 β activation is most likely very important in candidiasis.

Reviewer #2 (Remarks to the Author):

Kan et al. report on a concept that helps to understand why premature infants are more susceptible to bacterial and fungal infection. They describe actually two mechanisms, one specific for a defect in *Candida*-specific host defense, namely a defect in dectin-1-MALT signalling, and a general defect in preterm monocytes leading to unresponsiveness of cytokines by a diversity of PRR ligands explained by a general metabolic paralysis; impaired glycolysis, oxidative phosphorylation and β -oxidation. They shown novel and relevant data which fits Nature Communications, however the presentation of the data makes it unclear for the reader to dissect whether these are two separate defects going on in preterm infants. There are no experiments designed to link these two defects in order to make this clear and indeed translatable for the clinics.

Comments

The authors should point out that the studies performed with *Candida* and dissecting the pathway of a cytokine defect in response to *Candida* and curdlan as a dectin-1 ligand are done in a cell that has a general defect in metabolism making it already unresponsive to a wide variety of stimuli including TLR ligand such as LPS.

I suggest the authors start with the concept of complete metabolic paralysis (indeed mimicking LPS tolerance). This is an important observation.

The authors should measure lactate in the supernatants where cytokines are measured as a

functional readout for impaired glycolysis since they now have only one seahorse exp. with preterm monocytes showing a functional defect in addition to transcriptome profiling. The lactate should correlate with the capability to produce cytokines if the hypothesis of defective glycolysis is true in the preterm cells.

The section that follows should be starting with the remark that all exp. to study dectin-1 signalling are performed on the background of this general defect. However, it is still a nice observation. Why? Because if you would repair the metabolic (cytokine defect, tolerance status) with IFN γ the preterm cell could still be left with a specific dectin-1 defect making them more susceptible to Candida.

This is also the rationale and experiments lacking.

A few things could be going on here:

1. The defects are two separate defects (very relevant!) but they are not connected.
2. The initial defect in β -glucan makes the cells unable to overcome the tolerance state (Novakovic et al. Cell 2016). In this way the mechanisms could be linked, this should at least be discussed.
3. The impaired metabolic status leads to a MALT1 deficient signalling or the other way around. These hypothesis could be explored in preterm cells but also conceptually in healthy monocytes. Preterm cells : treat them in-vitro with IFN γ (as the authors already mention in the discussion) and see if the general cytokine production and glycolysis (as measured by lactate) improve. If so they can then test in this setting also dectin-1 signalling that would give the answer if it is still impaired on curdlan or not and would thus provide an answer whether the defects are linked. Another approach is to make monocytes tolerant with LPS and then test dectin-1_MALT signalling to see whether impaired glycolysis is linked with MALT1 deficiency. And vice versa to make a cell MALT deficient (with a blocker) and test for readout of impaired glycolysis in response to TLR ligands (measure cytokines and lactate). In my view only these exp. Could provide a hint towards the observations: whether the authors describe 2 defects that are not linked and thus two separate relevant stories or that they are interconnected.

Dectin-1 is a receptor for β -glucan, and purified β -glucan will by itself not induce significant proinflammatory cytokines in a human monocyte. To test true dectin-1 signalling effects another functional assay would be to use β -glucan purified from Candida with or without Pam3Cys to see if there is synergisms for TNF. If the experiments are not performed the authors should at least discuss that cytokine production in monocytes in response to a dectin-1 stimulus is not straightforward, and has also been proposed to only induce IL-1Ra.

Moreover, if there is still ng of production of IL-1 β one could still argue whether these cells cannot be inflammatory and result in an innate immune defect, this should also be discussed.

Minor:

The authors should discuss the observation of trained immunity (β -glucan) in the context of their observations (Cheng et al Science 2014)

“immune activation”

Please show the curdlan-induced cytokine deficiency in preterms in Figure 3.

Reviewer #3 (Remarks to the Author):

In this work, Dr. Kan and Colleagues investigated responses to Candida species using transcriptomic, metabolic and polysome profiling approaches. They have included a significant body of work and are to be congratulated for their efforts. They conclude they provide the first evidence of a role for cellular energy metabolism regulating neonatal innate immune responses during ontogeny, in a gestational age-dependent manner. I have a few comments.

The demographics of the groups were not given. The most concerning limitation of this work is that the group of preterm infants studied (<33 weeks) are very frequently exposed to antenatal

steroids with the goal of reducing respiratory distress syndrome and other post-natal conditions. In addition to antenatal betamethasone exposure for a goal duration of 48 hours prior to delivery, fetuses of mothers with preterm labor or with indications for preterm delivery are often exposed to additional immunomodulating and potentially metabomodulating substances such as high-dose magnesium (neuroprotection with pre-eclampsia) and tocolytics (Ca channel blockers, prostaglandin inhibitors). A mother with a term fetus would not be expected to receive these medications, nor would a healthy adult used for comparison studies. The presence of labor (contractions, cervical change) and the resultant vaginal delivery can modify cord blood cellular activation states as compared to C-sections without labor. Depending on the local c-section rate, many of the term infants may have been delivered vaginally while many of the preterm infants may have been delivered by c-section and without labor (preeclampsia). Furthermore, the group of preterm infants studied was <33 weeks of age, which without revealing what infants were studied means that may include newborns from a very wide range of development (23-32 weeks). These conditions would be expected to modify many of the parameters the authors studied and thus make the large amount of data difficult to interpret.

Newborn cord blood is acquired immediately after birth from the placental end of the umbilical cord after it has been clamped and cut to free the infant. Comparisons of newborn cord blood to that of peripheral blood from healthy adults are commonplace in the literature but are difficult to interpret. The process of birth including labor and delivery for the "healthy" infant cannot reasonably be considered a "state of health" or a "resting state" akin to a healthy adult. Cord blood is drawn after a newborn comes into the world through a process that could never be confused with a resting state of health in adult. The practice of collecting and studying cord blood largely results from the blood volume restrictions on newborns after birth, particularly in the extremely low birth weight infant (<1 kg), which may have total circulating blood volume of <50mL. The authors should clarify the limits of such a comparison to the reader.

Minor points

The incidence of Candida infection in ELBW has fallen dramatically during the last 15 years with a commensurate reduction in H2 blockers, cephalosporins, steroids, as well as the use of diflucan prophylaxis and improved line care bundles. There are inflammasome-independent ways to generate IL-1b from its proform including neutrophil serine proteases (PMC3482477). PMNs and not monocytes and macs may be the predominant source of IL-1b (PMID: 23209417). The description for the antibody used to identify pro-IL-1 beta did not suggest it could distinguish pro- and mature forms (BioLegend, #508208). It is unclear what effects prior freezing may have on glucose uptake in monocytes.

Throughout the manuscript, there are grammatical errors that are distracting. For example, the first sentence of the introduction reads "Infection is the leading causes of neonatal mortality worldwide...". The last sentence of the first paragraph in the introduction reads "Despite that the major clinical impact of these functional immune changes, we lack a molecular understanding of these responses are regulated during ontogeny." More examples later in draft, "The dectin-1 protein was comparable expressed between all 3 age groups (fig. 5E)", "Indeed, iimmune activation triggers...".

Reviewers' comments

Reviewer #1

*General comments: The present manuscript investigates the mechanisms that are responsible for the defective antimicrobial responses in premature infants. The focus is the invasive infection with *Candida albicans*, the most common human fungal pathogen that causes severe infections in premature newborn. The subject of the study is very relevant, the approach innovative, and the experiments are well performed. The authors present important new information regarding a role for cellular metabolism and mTOR. However, there are also several questions remaining.*

1. The authors use as last step of monocyte purification a positive selection using CD14 selection kit. The interaction of monocytes with the CD14 antibody can activate cells. Can the authors confirm the most important metabolic findings using a negative selection kit?

****Reply:** Although the possibility that altering the biology of monocytes by engaging CD14 during the cell isolation process is a theoretical concern, in our experience this has not seemed to be a real problem. In a previous publication, we report diminished PRR responses in preterm monocytes, including diminished LPS responses in both positively and negatively selected (untouched) monocytes (Sharma et al. 2015). In this study, we also confirmed some of the main findings without positively selecting monocytes. For example, low cytokine responses to dectin in preterm monocytes were measured in mononuclear cells (Fig. 1B) and in unselected monocytes, gating on CD14-expressing cells by flow cytometry (Fig. 1D). Although our array data was generated on CD14-purified monocytes to minimize contamination from other cells, the dissociation between gene and protein expression has also been confirmed in unselected monocytes (new Supp Fig. 2 provided). As for the functional metabolic experiments, results were confirmed in both CD14-positively selected (Fig. 7B, C) and unselected (Fig. 7D; lactate; Supp Fig. 8; glucose uptake) cells. mTOR experiments were also confirmed both in CD14-positively selected (Fig. 7G,H) and unselected (Supp Fig. 9) cells. Furthermore, all experiments were similarly carried out in adult, term and preterm cells. We feel that it is unlikely that activation of cells during the purification process may only affect preterm, but not adult or term neonatal cells. Altogether, the main have been validated using multiple independent approaches making use of positively and negatively selected cells, and we are confident that the findings are not an artifact due to the activation (or blocking) of CD14 during the cell isolation process.

2. The authors report diminished glycolysis in monocytes of preterm infants. It would be important to report however the behaviour of other important pathways such as oxidative phosphorylation and beta-oxidation. The authors have using Seahorse technology, why do they not report also oxygen consumption rate?

****Reply:** We now report ATP-linked oxygen consumption in Supplemental Fig. 10. The reviewer is right in that oxygen consumption rate is measured in parallel to running the ECAR. However, the measure of mitochondrial function utilizes different inhibitors than the one used when measuring ECAR. We have not performed detailed testing of beta-oxidation.

It is important to point out that over 100 preterm cord blood samples were used in the final experiments presented in this paper, in addition to the preterm samples that were required to optimize experimental conditions for each assays (e.g. testing of signal sensitivity, cell number, for example for the polysome experiments). Some experiments necessitate replication up to 20+ times due to the typical variability in humans. In this article, we initially tried to prioritize experiments that had the greatest potential for discovery. Now with the retrospective eye, it becomes a logical next question to ask what sources of energy are utilized by preterm cells other than glycolysis. This will likely require more than just assaying beta-oxidation on the seahorse in order to substantially gain knowledge on this question. In the future, we hope to optimize experiments to assay beta-oxidation metabolism together with non-targeted metabolomics (with our collaborators expert in this area). However, this is outside the scope of this article.

3. Differential roles of glycolysis and oxidative phosphorylation for cytokine production and phagocytosis has been recently reported for adult monocyte s (Lachmandas et al, Nature Microbiol 2016). How do preterm monocytes behave in that respect?

****Reply:** This is an interesting question. We have added this key reference to the result section, page 6. We also performed additional analyses focused in the re-wiring signature pathways described in the Lachmandas article. Our new results (page 6, result section, fourth paragraph) show that both glycolytic and ox phos pathways are down-regulated in preterm cells. Interestingly, we present new data and discuss an up-regulation of PPAR-gamma and its target metabolic genes in preterm cells (see newly added supplemental fig. 10 and Supp fig. 11). PPAR-gamma has been shown to promote phagocytosis and limit pro-inflammatory responses. This suggest that preterm monocytes behave distinctly than described by Lachmandas in PRR-rewired adult cells.

According to the article by Lachmandas et al, the differential role of glycolysis and oxidative phosphorylation for cytokine production and phagocytosis appears to be pathogen-dependent. For *Candida*, we tested whether blocking of glycolysis or oxidative phosphorylation affected basal or LPS-primed (24h, as used in reference cited) phagocytosis responses in a small number (n=3) of adult monocyte samples and did not detect a major effect with 2-DG or rotenone. We conclude that phagocytosis of *Candida* is generally not dependent on ox phos as maybe for stimulation of cells by certain PRR (e.g. P3C) as in the article by Lachmandas.

4. The authors briefly present the differential recognition of yeasts and hyphae from *Candida albicans* by PRRs. The recognition of hyphae is also done by TLR2 and TLR4, as described by the authors, but these are not the main receptors. The main recognition receptors for hyphae are still of the C-type lectin receptor family, with important roles for dectin-1 and dectin-2.

****Reply:** Thank you for the precision. We have amended our introduction to reflect this.

5. The processing and activation of IL-1 β is actually more complex than described by the authors. In addition of cysteine proteases (such as caspase-1 and caspase-8), a major pathway of activation is through cleavage by serine proteases (e.g. PR3, cathepsin G, etc) especially from neutrophils. As neutrophils are crucial cells for anti-*Candida* host defense, this inflammasome-independent pathway of IL-1 β activation is most likely very important in candidiasis.

****Reply:** This has now been clarified in the discussion, limitation section, page 9.

Reviewer #2

General comments: Kan et al. report on a concept that helps to understand why premature infants are more susceptible to bacterial and fungal infection. They describe actually two mechanisms, one specific for a defect in *Candida*-specific host defense, namely a defect in dectin-1-MALT signalling, and a general defect in preterm monocytes leading to unresponsiveness of cytokines by a diversity of PRR ligands explained by a general metabolic paralysis; impaired glycolysis, oxidative phosphorylation and β -oxidation. They show novel and relevant data which fits Nature Communications,

1. The presentation of the data makes it unclear for the reader to dissect whether these are two separate defects going on in preterm infants. There are no experiments designed to link these two defects in order to make this clear and indeed translatable for the clinics.

****Reply:** New data have been added in Figure 8 to see whether the dectin-1 signalling defect and monocyte non responsiveness are linked. Essentially, these data show that blocking glycolysis in human monocytes results in a gradual loss of MALT1 protein expression over time (Fig. 8A). Conversely, blocking of MALT1 does not impair glycolysis, as shown by lactate levels both at rest (unstimulated cells) and after LPS stimulation (Fig. 8B). These indicate that the lack of MALT1 protein expression is downstream of the glycolytic switch rather than the other way around.

2. The authors should point out that the studies performed with *Candida* and dissecting the pathway of a cytokine defect in response to *Candida* and curdlan as a dectin-1 ligand are done in a cell that has a general defect in metabolism making it already unresponsive to a wide variety of stimuli including TLR ligand such as LPS.

****Reply:** This has now been included in the discussion: (Page 8, second paragraph starting with sentence: "Our data have clinical relevance....", with new reference 46.

3. The authors should measure lactate in the supernatants where cytokines are measured as a functional readout for impaired glycolysis since they now have only one seahorse exp. with preterm monocytes showing a functional defect in addition to transcriptome profiling. The lactate should correlate with the capability to produce cytokines if the hypothesis of defective glycolysis is true in the preterm cells.

****Reply:** The relationship between cytokine response and lactate is indirectly shown in Fig. 7A and D. We have added new data in Fig. 8C directly correlating cytokines and lactate levels in adult cells. Correlation of lactate with IL-1 β is imperfect and weaker than IL-6, possibly due to the dual activation required for secretion of IL-1 β resulting in kinetics effects that confound this association. In preterm cells, cytokine levels did not correlate with lactate at all likely due to very low cytokine and low lactate levels (graph immediately below), for these reasons a lack of correlation between lactate and cytokine measured at a single cross-section time point may not exclude an effect of defective glycolysis in preterm cells.

The section that follows should be starting with the remark that all exp. to study dectin-1 signalling are performed on the background of this general defect. However, it is still a nice observation. Why? Because if you would repair the metabolic (cytokine defect, tolerance status) with IFN γ the preterm cell could still be left with a specific dectin-1 defect making them more susceptible to Candida. This is also the rationale and experiments lacking.

A few things could be going on here:

1. The defects are two separate defects (very relevant!) but they are not connected.
2. The initial defect in β -glucan makes the cells unable to overcome the tolerance state (Novakovic et al. Cell 2016). In this way the mechanisms could be linked, this should at least be discussed.
3. The impaired metabolic status leads to a MALT1 deficient signalling or the other way around.

****Reply:** The new data provided in Fig. 8 suggest that the 2 defects are linked and that impaired metabolism leads to MALT1 deficient signalling rather than the other way around.

These hypotheses could be explored in preterm cells but also conceptually in healthy monocytes.

Preterm cells: treat them in-vitro with IFN γ (as the authors already mention in the discussion) and see if the general cytokine production and glycolysis (as measured by lactate) improve. IF so they can then test in this setting also dectin-1 signalling that would give the answer if it is still impaired on curdlan or not and would thus provide an answer whether the defects are linked. Another approach is to make monocytes tolerant with LPS and then test dectin-1_MALT signalling to see whether impaired glycolysis is linked with MALT1 deficiency. And vice versa to make a cell MALT deficient (with a blocker) and test for readout of impaired glycolysis in response to TLR ligands (measure cytokines and lactate). In my view only these exp. Could provide a hint towards the observations: whether the authors describe 2 defects that are not linked and thus two separate relevant stories or that they are interconnected.

****Reply:** The increase in glycolysis with interferon-gamma treatment has already been described previously (Cheng, 2016; figure 6). This reference has been added to the discussion (page 9).

We attempted the experiments suggested by the reviewer. As shown in figure immediately below we observed an increase in LPS responses with 1h interferon-gamma pre-treatment of adult mononuclear cells, although same pre-treatment conditions resulted in diminished *in vitro* responses to Candida (A). We suspect that this may be due to a rapid change in maturation of monocytes <24h in culture conditions that reduces responsiveness to yeast Candida. Also, as shown in figure/panel (B) below, monocytes cultured for 5 days (as described in Netea's high dose LPS tolerance protocol) resulted in further substantially reduction in both Candida responsiveness and IL-1 β production. For these reasons, we felt that the experiments proposed by Reviewer#2 were not going to answer the question adequately due to a limited relevance of the vitro model of IFN-g or tolerance on MALT1.

Instead, we have found another way to address the link between the MALT1 signaling defect and monocyte non-responsiveness as provided in newly added Fig. 8. We hope that these new data satisfactorily address this question.

*Dectin-1 is a receptor for β -glucan, and purified β -glucan will by itself not induce significant proinflammatory cytokines in a human monocyte. To test true dectin-1 signalling effects another functional assay would be to use β -glucan purified from *Candida* with or without Pam3Cys to see if there is synergisms for TNF. If the experiments are not performed the authors should at least discuss that cytokine production in monocytes in response to a dectin-1 stimulus is not straightforward, and has also been proposed to only induce IL-1Ra. Moreover, if there is still ng of production of IL-1 β one could still argue whether these cells cannot be inflammatory and result in an innate immune defect, this should also be discussed.*

****Reply:** In a previous article (Sharma, 2015), we demonstrate that TLR-induced IL-1b responses are broadly deficient in preterm monocytes. In the present article, whole *Candida* induced a robust cytokine signal in adult cells, but not in preterm cells. If we understand the reviewer's comment correctly, he/she is wondering whether the lack of *Candida* recognition in preterm cells may be due to a lack of TLR1/2 synergism, although this is not suggested by our experiment in Figure 2A where response to *Candida* is blocked using a dectin-1 antibody.

The reviewer is probably referring to this paper: (<https://www.ncbi.nlm.nih.gov/pmc/articles/PMC5932370/>) which states "Of note, β -glucan/dectin-1 interaction per se is sufficient to induce the phagocytosis and production of ROS. In contrast, the production of inflammatory cytokines seems to be dependent on the cooperation between dectin-1 and toll-like receptors (TLRs)" citing 3 papers (PMID: 15304394, 12719479, and 12719478). These three papers used zymosan as a model, and showed that TLR and dectin-1 signalling is required for cytokine production by either depleting zymosan (so it only binds TLR, and not dectin), or using dectin/tlr knockout mice. We also note that they only looked at TNF-a as a readout for inflammatory cytokine production. In contrast, others have reported robust IL-1b production with dectin-1 stimulation alone via curdlan (PMID: 28736555, 20421639, 20976143, 25063877). Therefore, the literature is divided on this question (PMID: 26840954).

Minor: The authors should discuss the observation of trained immunity (β -glucan) in the context of their observations (Cheng et al Science 2014) "immune activation"

****Reply:** A brief sentence has now been included about β -glucan training, and the potential limitation of using this approach in preterm cells lacking functional dectin responses (page 9).

Please show the curdlan-induced cytokine deficiency in preterms in Figure 3.

****Reply:** This data has been added in supplemental figure 2.

Reviewer #3

*In this work, Dr. Kan and Colleagues investigated responses to *Candida* species using transcriptomic, metabolic and polysome profiling approaches. They have included a significant body of work and are to be congratulated for their efforts. They conclude they provide the first evidence of a role for cellular energy metabolism regulating neonatal innate immune responses during ontogeny, in a gestational age-dependent manner. I have a few comments.*

The demographics of the groups were not given. The most concerning limitation of this work is that the group of preterm infants studied (<33 weeks) are very frequently exposed to antenatal steroids with the goal of reducing respiratory distress syndrome and other post-natal conditions. In addition to antenatal betamethasone exposure for a goal duration of 48 hours prior to delivery, fetuses of mothers with preterm labor or with indications for preterm delivery are often exposed to additional immunomodulating and potentially metabomodulating substances such as high-dose magnesium (neuroprotection with pre-eclampsia) and tocolytics (Ca channel blockers, prostaglandin inhibitors). A mother with a term fetus would not be expected to receive these medications, nor would a healthy adult used for comparison studies. The presence of labor (contractions, cervical change) and the resultant vaginal delivery can modify cord blood cellular activation states as compared to C-sections without labor. Depending on the local c-section rate, many of the term infants may have been delivered vaginally while many of the preterm infants may have been delivered by c-section and without labor (preeclampsia). Furthermore, the group of preterm infants studied was <33 weeks of age, which without revealing what infants were studied means that may include newborns from a very wide range of development (23-32 weeks). These conditions would be expected to modify many of the parameters the authors studied and thus make the large amount of data difficult to interpret.

****Reply:** Demographics is now included for all preterm samples included in the study, as a supplemental data file. To preserve the confidentiality of the study participants and due to restrictions imposed by our institutional ethics committee, we are only allowed to present aggregated, not individual-level clinical characteristics.

We thank the reviewer for this suggestion and agree that the addition of clinical information on the more heterogeneous preterm group will greatly facilitate interpretation of the findings. While clinical characteristics of subjects may influence immune responses to *Candida*, the vast majority of responses from preterm subjects below 33 weeks are uniformly weak (see Figure 1B) and due to the relatively small sample size we are unable to provide detailed co-variate analyses.

Newborn cord blood is acquired immediately after birth from the placental end of the umbilical cord after it has been clamped and cut to free the infant. Comparisons of newborn cord blood to that of peripheral blood from healthy adults are commonplace in the literature but are difficult to interpret. The process of birth including labor and delivery for the "healthy" infant cannot reasonably be considered a "state of health" or a "resting state" akin to a healthy adult. Cord blood is drawn after a newborn comes into the world through a process that could never be confused with a resting state of health in adult. The practice of collecting and studying cord blood largely results from the blood volume restrictions on newborns after birth, particularly in the extremely low birth weight infant (<1 kg), which may have total circulating blood volume of <50mL. The authors should clarify the limits of such a comparison to the reader.

****Reply:** We have added this limitation in the discussion of the paper.

Minor points

The incidence of Candida infection in ELBWs has fallen dramatically during the last 15 years with a commensurate reduction in H2 blockers, cephalosporins, steroids, as well as the use of diflucan prophylaxis and improved line care bundles.

****Reply:** We added a reference to the decreasing incidence of neonatal infections due to Candida in the introduction.

There are inflammasome-independent ways to generate IL-1b from its proform including neutrophil serine proteases (PMC3482477). PMNs and not monocytes and macs may be the predominant source of IL-1b (PMID: 23209417). The description for the antibody used to identify pro-IL-1 beta did not suggest it could distinguish pro- and mature forms (BioLegend, #508208). It is unclear what effects prior freezing may have on glucose uptake in monocytes.

****Reply:** We have added a paragraph in the discussion to point out the role of neutrophils during infection. This study focused on monocytes in order to study cell intrinsic differences instead of global immune alterations which may be partially caused by differences in immune cell subsets. We agree that examination of other immune cells as well as whole blood studies would help translate our findings into the clinic.

Throughout the manuscript, there are grammatical errors that are distracting. For example, the first sentence of the introduction reads "Infection is the leading causes of neonatal mortality worldwide...". The last sentence of the first paragraph in the introduction reads "Despite that the major clinical impact of these functional immune changes, we lack a molecular understanding of these responses are regulated during ontogeny." More examples later in draft, "The dectin-1 protein was comparable expressed between all 3 age groups (fig. 5E)", "Indeed, iimmune activation triggers...".

****Reply:** We have gone through extensive revisions of the manuscript, also correcting minor errors from the previous submission. We hope that the new manuscript is exempt from grammatical errors.

Reviewers' comments:

Reviewer #1 (Remarks to the Author):

The authors answered properly my comments.

Reviewer #2 (Remarks to the Author):

The authors responses have clarified the message and data of the ms.

Reviewer #3 (Remarks to the Author):

The author's responses and changes to the manuscript are very much appreciated. The statement to readers describing the small size and fragility of the patients studied, including their limited blood volume, is appreciated and underscores why this population is important to study. Since the blood volume was deemed prohibitive for ethical and safety reasons as a peripheral draw, inclusion of the cord blood volume required (not what was drawn) in the methods would be meaningful. With respect, would the authors say the studies they have performed are technically impossible in mice due to blood volume restrictions? The declaration that cord blood "remains a suitable alternative" immediately after saying it "may not adequately represent systemic immune responses" to me still articulates the notion that it is ok to use cord blood as a surrogate for peripheral blood in this population and it is justified because it is unethical to draw the volume needed to perform the studies. Perhaps the authors would be given an opportunity to expand a little bit more in the discussion on what specific aspects of their study are supported by the refs they cite (58-60)? eg. Monocyte MHCII expression is similar between cord and peripheral blood from preterm infants (Marchant et al, Azizia et al.) rather than a broad immunologic stroke of "peripheral blood responses are comparably low (refs 58-60)."

Response to reviewers' comments:

Reviewer #3

The author's responses and changes to the manuscript are very much appreciated. The statement to readers describing the small size and fragility of the patients studied, including their limited blood volume, is appreciated and underscores why this population is important to study. Since the blood volume was deemed prohibitive for ethical and safety reasons as a peripheral draw, inclusion of the cord blood volume required (not what was drawn) in the methods would be meaningful. With respect, would the authors say the studies they have performed are technically impossible in mice due to blood volume restrictions? The declaration that cord blood "remains a suitable alternative" immediately after saying it "may not adequately represent systemic immune responses" to me still articulates the notion that it is ok to use cord blood as a surrogate for peripheral blood in this population and it is justified because it is unethical to draw the volume needed to perform the studies. Perhaps the authors would be given an opportunity to expand a little bit more in the discussion on what specific aspects of their study are supported by the refs they cite (58-60)? eg. Monocyte MHCII expression is similar between cord and peripheral blood from preterm infants (Marchant et al, Azizia et al.) rather than a broad immunologic stroke of "peripheral blood responses are comparably low (refs 58-60)."

****Reply:** We have revised the limitation paragraph of the discussion to emphasize the functional characteristics of preterm cord blood that have been reproduced on peripheral blood. In doing so, we found an erroneous citation (Zhang et al.) that now has correctly been replaced to Shen SC et al. (Ref. #63). We also indicate the blood volumes that were generally required in our experiments. The experiment that required the largest blood volume was the polysome profiling where we needed to use at least 5 mL of cord blood (depending of course on the donor) to detect sufficient gene expression in the low disome fractions. The strict minimal blood volume that was isolated monocytes from was 1 mL although we most often had to use much more, which is ethically extremely difficult to justify with peripheral blood. Regarding the question of whether these experiments are possible in mice. While we are not expert in this field, immunological studies in mice are not as limited to blood sampling. The use of bone-marrow-derived macrophages (BMDM, not possible in humans), splenocytes, *in vivo* experiments and combining samples from (syngeneic) mice allows circumventing many of the experimental issues associated with limited blood volumes.

Regarding the manuscript results, we made the following corrections:

- 1) Supplementary Figure 8, the statistics and methods were incorrect – the difference between groups was tested at baseline (unstimulated) using a cumulative distribution Kolmogorov-Smirnov test for non-parametric data. The figure legend has been changed accordingly.
- 2) Supplemental Figure 14, panel B, the error bars for the term and adult data represent SD - this info has been added.
- 3) While this article was under review, we worked to address whether there could be a difference in glycolytic capacity between term and adult monocytes. New data has been added to figure 7C, confirming the same trend as shown previously. The interpretation of this result remains unchanged. The data is now reported as non-(protein) normalized values with same number of cells tested in all conditions (protein normalization can be a problem due to interference of the test medium itself, as repeatedly reported by other authors).

In addition, we re-checked the references have made several minor language and formatting edits to the manuscript for clarity, and to comply with the journal requirements. None of these latter changes affect the content and interpretation of the data in any substantial way.

REVIEWERS' COMMENTS:

Reviewer #3 (Remarks to the Author):

I sincerely thank the authors for their contributions and their responses.